# Determinants of sensitivity to HER2-targeted antibody drug conjugates in urothelial cancer

Ziyu Chen[1,2,14], Xinran Tang[1,2,14], Jordan E. Eichholz[3], Andrew Mcpherson[3,4], Jasmine Thomas[5], Karan Nagar[5], Naryan Rustgi[1], John R. Christin[6], Fengshen Kuo[1,5], Sizhi Gao[1], Hui Jiang[1,5], Jiaqian Luo[1,2], Irina Ostrovnaya[7], Merve Basar[8], Eugene Pietzak[9], Jonathan A. Coleman[9], Michael F. Berger[8,10], Elisa de Stanchina[11], Sohrab P. Shah[3,4], Neeman Mohibullah[12], David H. Aggen[13], Jonathan E. Rosenberg[13], Sarat Chandarlapaty[1], Michael M. Shen[6], Hikmat Al-Ahmadie[8], Gopa Iyer[13], Kwanghee Kim[5] & David B. Solit[1,10,13]✉

HER2, encoded by the *ERBB2* gene, is a receptor tyrosine kinase frequently activated in human cancers via gene amplification, mutation, and/or protein overexpression. In an analysis of 42,415 prospectively analyzed solid tumors, we show that 14.5% of urothelial cancers (n = 295/2,035) have oncogenic or likely oncogenic *ERBB2* alterations (6.7% *ERBB2* mutation, 6.3% amplification of wildtype *ERBB2*, and 1.5% concurrent mutation and amplification). Discordance of *ERBB2* mutational status between primary and metastatic disease sites is common in patients with urothelial cancer as is discordance of *ERBB2* mutational status between patient-derived organoid/xenograft models and the tumors from which they were derived. In patient-derived urothelial cancer models, the HER2-targeted antibody-drug conjugate (ADC) trastuzumab deruxtecan is significantly more effective than the HER kinase inhibitor neratinib. In a real-world cohort of patients with urothelial cancer treated with trastuzumab deruxtecan, co-mutation and amplification of *ERBB2* is associated with exceptional clinical response. Our data support expanded clinical trials of HER2-targeted ADCs for urothelial cancers with low HER2 expression, the clinical testing of HER2 ADCs with alternative cytotoxic payloads, and the development of functional precision oncology platforms capable of assessing payload sensitivity pre-treatment as a guide to individualized therapy selection.

Treatment options for patients with bladder and upper tract urothelial cancer have expanded significantly over the past several years. Successful clinical trials have resulted in FDA approval of novel therapies spanning three mechanistically distinct drug classes: immune checkpoint inhibitors (anti-PD-1/PD-L1 antibodies), fibroblast growth factor receptor 3 (FGFR3) kinase inhibitors, and antibody drug conjugates (ADCs) targeting Nectin-4 and HER2[1–11]. HER2, which is encoded by the *ERBB2* gene, is a non-ligand-binding member of the epidermal growth factor family of receptor tyrosine kinases[12]. Despite a high prevalence of HER2 alterations in urothelial cancer, clinical trials of older HER2-directed therapies including the monoclonal antibody trastuzumab and HER2-selective kinase

inhibitors such as neratinib reported disappointing results in patients with advanced urothelial carcinomas[13–16].

Trastuzumab deruxtecan (T-DXd) is a second-generation HER2-targeted ADC with a tetrapeptide-based cleavable linker and a topoisomerase I inhibitor payload. T-DXd has demonstrated clinical activity in patients whose tumors were resistant to older HER2-targeted therapies including trastuzumab-resistant HER2-positive breast cancers[17,18]. Recent clinical results from two T-DXd basket trials (DESTINY-PanTumor01 and DESTINY PanTumor02) have also reported compelling clinical activity pan-cancer, including in patients with HER2 positive and mutant urothelial cancers. Most notably, 39% of patients with HER2 positive urothelial cancer enrolled on the DESTINY PanTumor02 study had a partial response, and 87% stable disease or better[19]. These results were sufficiently compelling to justify a tumor agnostic accelerated FDA approval for T-DXd for the treatment of unresectable or metastatic HER2 (3+ by IHC)-positive solid tumors, including urothelial cancers. However, the need for HER2 amplification or mutation to achieve T-DXd clinical benefit has been questioned based on the compelling clinical activity of T-DXd in patients with HER2-low (IHC 1+ or 2+/ISH-negative)[18] and even HER2 "ultra-low" (IHC 0 with membrane staining)[20] breast cancers. Together with the mixed clinical results of HER2-targeted therapies in patients with urothelial cancer there remains uncertainty as to which subsets of patients with urothelial cancer should be the focus of future clinical trials of novel HER2-directed therapies.

To address these questions, we analyzed tumor sequencing data generated within the context of a prospective, institution scale genomic profiling initiative to define the landscape of HER2 alterations in bladder and upper tract urothelial cancers, and their association with patient demographic and pathologic features. We then leveraged a newly generated biobank of patient-derived organoid and xenograft models of urothelial cancer to study the relationship between HER2 mutational status, HER2 expression, HER2 oncogenic dependence, and sensitivity to HER2 targeted ADCs in a urothelial cancer context. Finally, to infer the timing at which ERBB2 amplification arose during disease pathogenesis and the potential clinical impact of genomic heterogeneity on HER2-targeted therapy response, patient-derived urothelial cancer organoid and xenograft models were characterized at the single-cell level for HER2 genomic and expression heterogeneity and sensitivity to HER2-targeted kinase inhibitors and ADCs.

In this work, we observe significant ERBB2 mutational heterogeneity in patients with urothelial cancer, including discordance in ERBB2 mutational status between primary and metastatic tumors sites, as well as discordance of a subset of patient-derived models and the urothelial tumors from which they were derived. In urothelial cancer PDX models, T-DXd demonstrates superior anti-tumor activity compared to the HER2 kinase inhibitor, neratinib. Retrospective analyses of a real-world cohort of urothelial cancer patients treated with T-DXd identified co-mutation and amplification of ERBB2 in patients with exceptional response to T-DXd.

## Results

### The genomic landscape of ERBB2 mutated/amplified urothelial cancer

To define the prevalence of ERBB2 mutation and gene amplification in urothelial cancer and its association with patient demographic and pathologic features, we analyzed tumor genomic profiling data from an institutional scale, prospective tumor profiling initiative[21]. In a cohort of 42,415 prospectively analyzed solid tumors, 14.5% of urothelial cancers (n = 295/2,035) had ERBB2 alterations. Among common solid tumors, urothelial cancers had the highest frequency of oncogenic and likely oncogenic ERBB2 mutations, with ERBB2 mutations identified in 8.2% (inclusive of both bladder and upper tract urothelial cancer (UTUC) primary sites), 18.2% of which had concurrent ERBB2 amplification. Additionally, amplification of wildtype ERBB2 was observed in 6.3% (Fig. 1A, and Supplementary Fig. 1A).

ERBB2 alterations were more common in tumors with a bladder primary site versus tumors arising in the ureter or renal pelvis (22.0% for bladder vs 16.5% for upper tract urothelial cancers (UTUC), p = 0.042, Supplementary Fig. 1B). In urothelial cancer, ERBB2 mutations were predominantly localized to the extracellular domain (52.4%, 144/275) with mutations involving the oncogenic S310F/Y allele most common (6.8% of bladder and 3.1% of UTUC tumors harbored a S310F/Y mutation, Fig. 1B, and Supplementary Fig. 1C)[22,23]. This high prevalence of ERBB2 S310F/Y mutation in urothelial cancers was likely driven by its association with an APOBEC mutagenic process (Supplementary Fig. 1D)[24]. While there was no correlation between ERBB2 mutational status and sex, race or smoking status, ERBB2 alterations were more frequent in higher grade and stage tumors (q = 0.001), as well as in urothelial cancers with micropapillary histology (q = 0.001, Supplementary Fig. 2A–C)[13,25,26]. For 30 ERBB2 altered patients in the cohort, tumor genomic profiling data was available for both primary and metastatic tumor sites. ERBB2 alterations were frequently discordant (12/30, 40%) in patient matched primary and metastatic tumor pairs, with 5 tumors having ERBB2 alterations exclusive to the primary tumor and 7 ERBB2 alterations exclusive to the metastasis[27]. Phylogenic analysis of primary and metastatic tumors from two representative patients are highlighted in Supplementary Fig. 2D, confirming that such ERBB2 discordant primary-metastasis pairs were clonally related.

For older HER2-targeted therapies such as trastuzumab and kinase inhibitors, there is a strong correlation between ERBB2 gene amplification and treatment response[28]. Given the disappointing results with trastuzumab and the kinase inhibitor neratinib in patients with urothelial cancer[16,29], we asked whether urothelial tumors classified clinically as ERBB2 amplified had levels of gene amplification similar to that of cancers for which trastuzumab and HER2 kinase inhibitors are standard care. Integer ERBB2 copy number inferred from MSK-IMPACT data revealed that ERBB2 amplified esophagogastric cancers had the highest median ERBB2 copy number (26, inter-quartile range IQR:10-42.75), followed by colorectal and breast cancers (22, IQR: 9-36 and 21, IQR:12-36, respectively), with median ERBB2 copy number lower in urothelial cancers (17.5, IQR:8.25-29.25) (Fig. 1C). Tumors with high levels of ERBB2 amplification (>20 copies), were also significantly more common in patients with breast (7.06%) and esophageal cancers (10%) than urothelial cancers (2.93%) (p < 0.001, Supplementary Fig. 1E).

As the pattern of co-mutation has been demonstrated to influence sensitivity to HER2-targeted therapies in other cancer types[30–32], we defined the patterns of mutational co-occurrence in ERBB2 mutant and amplified urothelial cancers. Several notable differences in the pattern of mutational co-alteration in ERBB2 mutant versus amplified tumors were observed that could impact HER2-oncogenic dependence and sensitivity to HER2 targeted therapies, in particular antibody drug conjugates. Whereas both ERBB2 mutation and amplification were largely mutually exclusive with oncogenic mutations in FGFR3, only ERBB2 amplification was significantly co-associated with TP53 and ERCC2 alterations, whereas ARID1A and TERT alterations were enriched in tumors with ERBB2 mutations (Supplementary Fig. 3A). We also compared tumor mutational burden (TMB) and fraction genome altered (FGA) in urothelial cancers with and without ERBB2 alterations. ERBB2 altered urothelial cancers had a significantly higher median TMB (WT: 8.2 vs. AMP: 11.4, p < 0.001; WT: 8.2 vs. MUT: 11.4, p < 0.001). However, FGA was not significantly different between ERBB2 mutated and wildtype tumors, with both having a median FGA significantly less than ERBB2 amplified tumors (WT: 0.16 vs. AMP: 0.29, p < 0.001; WT: 0.16 vs. MUT: 0.18, p = 0.23, respectively, Supplementary Fig. 3B).

Finally, we performed an integrated analyses of DNA and RNA sequencing data generated by the Tumor Cancer Genome Atlas to assess the correlation between ERBB2 mutational status and mRNA expression and HER2 protein expression as assess by Reverse Phase Protein Array (RPPA)[33]. In comparison to breast cancers in which there

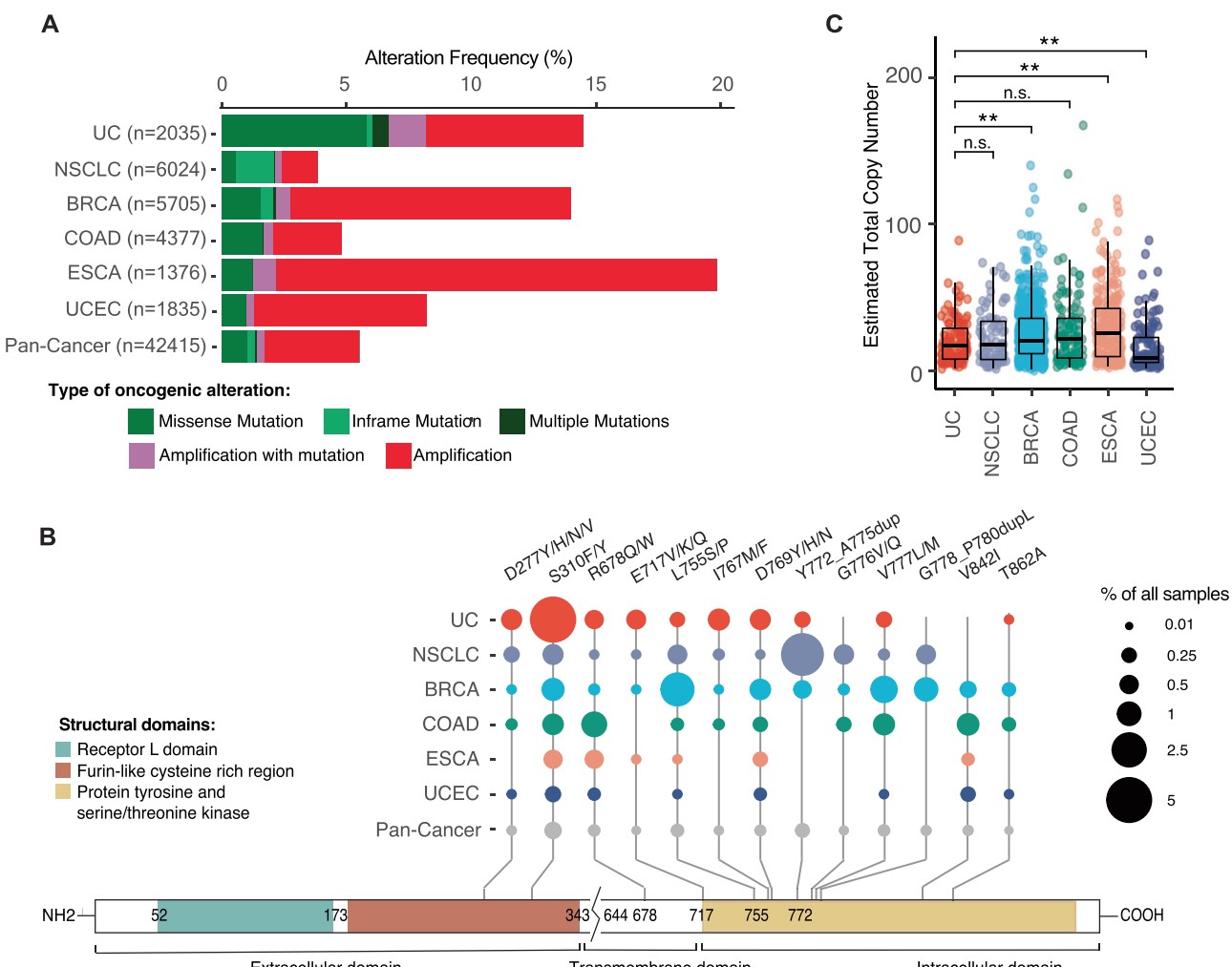

**Fig. 1 | The landscape of *ERBB2* alterations pan cancer. A** Prevalence of *ERBB2* mutation/amplification in patients with urothelial and other common cancer types in a cohort of 42,415 tumors analyzed using the MSK-IMPACT assay. UC: bladder and upper tract urothelial cancer, COAD: colorectal adenocarcinoma, BRCA: breast cancer, ESCA: esophagogastric cancer, UCEC: endometrial cancer, NSCLC: non-small cell lung cancer. **B** Lollipop plot of *ERBB2* hotpot mutations in common cancer types. The size of each circle denotes the proportion of each *ERBB2* variant as a percentage of all *ERBB2*-altered tumors per cancer type. Structural domains are depicted in the schematic below the plot. **C** Estimated *ERBB2* copy number based on FACETS analysis. The lower and upper hinges correspond to the first and third quartiles, and the middle line represents the median. The whiskers extend 1.5*IQR from the hinge. Significance denoted as ** $p < 0.01$, n.s. not significant. *P* values were determined by a two-sided Wilcoxon rank-sum test without adjustments. N numbers, median, and specific *p* values are provided in Source Data.

was a strong correlation between *ERBB2* mRNA and HER2 protein levels (cor = 0.79, p < 0.001), this association was weaker in urothelial cancers (cor = 0.72, p < 0.001) (Supplementary Fig. 3C). We also assessed for differences in transcriptional and/or immune signatures as a function of *ERBB2* mutational status. While there was a trend towards a higher proportion of *ERBB2* amplified tumors having a luminal transcriptional phenotype (Supplementary Fig. 3D)[34], this trend was not statistically significant. Immune deconvolution analysis demonstrated significant enrichment of NK CD56[bright] cells in the *ERBB2* amplified tumors and enrichment of Th17 cells in *ERBB2* mutated tumors suggesting that *ERBB2* alteration may be associated with a distinct immune landscape in patients with urothelial cancer (Supplementary Fig. 3E).

### Single-cell characterization of HER2-altered bladder cancer patient-derived models

A lack of HER2-mutated and amplified patient-derived models of urothelial cancer has been a major impediment to preclinical studies of novel HER2-targeted therapies for this disease. To address this limitation, we generated and analyzed a biobank of 45 patient-derived

xenograft or organoid models (PDX-O) that more accurately reflect the biologic heterogeneity of the human disease than older 2D cell lines (Fig. 2A, and Supplementary Table 1). Among the PDO models, 4 had *ERBB2* amplification only, 3 *ERBB2* mutation only, and one both *ERBB2* amplification and mutation. To confirm that the PDX-O models mirrored the genomic profile of the tumors from which they were derived, we performed targeted and/or whole exome sequencing of both the patient-derived model and the tumor from which it was derived. (Fig. 2A, B, and Supplementary Fig. 4A, B). While the somatic mutational profiles of the patient tumor/PDX-O pairs were largely concordant, we observed a high degree of *ERBB2* mutational discordance. For example, although 72 (61.5%) mutations, including an oncogenic *RB1* S249* mutation, were shared between the SMBO-109 PDO and the tumor from which it was derived, *ERBB2* amplification was present only in the organoid line but not in the tumor (Fig. 2A, B). In total, 46% (6 of 13) of the patient-matched tumor-PDX-O pairs with a *ERBB2* mutation or amplification detected in either, had discordance of *ERBB2* mutational status, results consistent with the high frequency of *ERBB2* mutational discordance observed in our analysis of primary–metastasis tumor pairs[27].

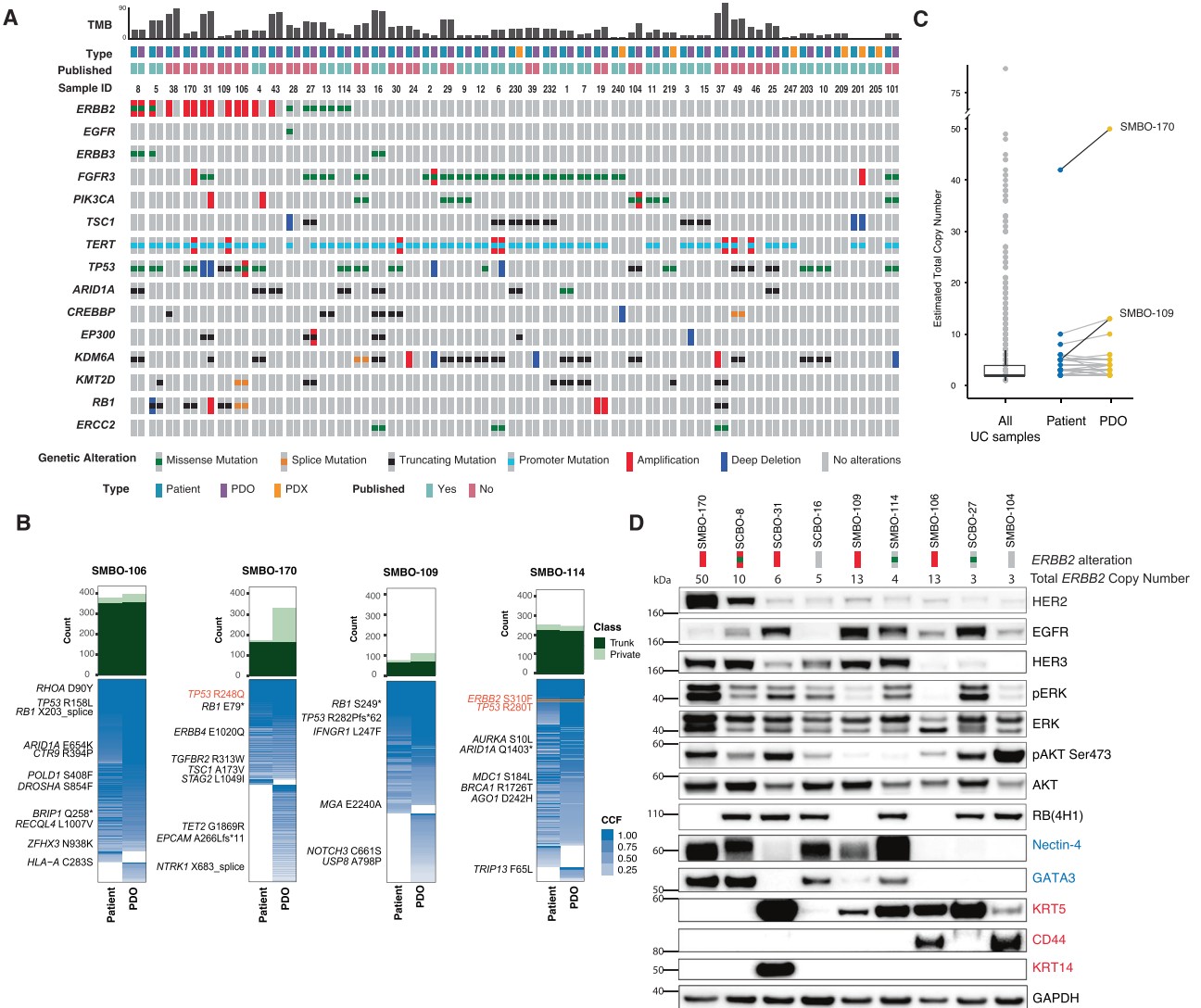

**Fig. 2 | Molecular characterization of patient derived organoid and xenograft models of urothelial cancer. A** Oncoprint of select genes for 45 patient derived xenograft or organoid (PDX-O) models of urothelial cancer. Tumor DNA was derived from PDX-O models and the tumors from which they were derived. Blood was used as a source of germline DNA. Mutational status of the patient tumors is shown on the left and for the patient-matched PDX-O on the right. **B** Phylogenic analysis of WES data from 4 urothelial cancer PDOs and the corresponding tumors from which they were derived. The number of mutations shared between the tumor and organoid is shown in dark green, and the number exclusive to either the tumor

or organoid in light green. In the respective phylogeny, cancer cell fraction is shown as shades of blue. **C** Estimated total *ERBB2* copy number of 24 patient-matched tumor/PDX-O pairs (right), with the overall MSK urothelial carcinoma cohort (left) as a comparator. **D** Immunoblot of select patient derived organoids demonstrating relative expression of HER2, EGFR, phosphorylated ERK (pERK, Thr202/Tyr204), ERK, phosphorylated AKT (Ser473), AKT, RB1, Nectin-4, GATA3, KRT5, CD44, KRT14 and GAPDH (as a loading control). Luminal markers are indicated in blue and basal markers in red. The samples were derived from the same experiment, and the blots were processed in parallel.

To quantify the variation in *ERBB2* amplification across urothelial cancer models and patient tumors, we next inferred integer *ERBB2* copy number from the next generation sequencing data. The estimated *ERBB2* total copy number was similar between the PDO models and the tumors from which they were derived, with the notable exception of SMBO-109 (*ERBB2* copy number of 13 for the PDO and 5 for the tumor, Fig. 2C). Finally, we quantitated the expression of HER2 as well as downstream mediators of HER2 signaling in a panel of 9 urothelial cancer PDOs. HER2 protein expression was highest in SMBO-170 and SCBO-8 (Fig. 2D), consistent with their higher levels of *ERBB2* amplification (an estimated 50 and 10 copies of *ERBB2* in SMBO-170 and SCBO-8, respectively). HER2 protein expression as quantitated by immunoblot was not, however, significantly elevated in models with lower levels of *ERBB2* copy number gain versus PDO models wildtype for *ERBB2*. These data suggest that the threshold employed clinically

for identifying urothelial tumors with *ERBB2* amplification likely overstates the frequency of amplification mediated HER2 protein overexpression and are consistent with the lower degree of correlation between *ERBB2* amplification and HER2 overexpression noted in urothelial versus breast cancers in the TCGA cohort (Supplementary Fig. 3C).

To characterize HER2 heterogeneity at single-cell resolution and to infer the timing at which *ERBB2* amplifications arose during clonal evolution, we performed single-cell DNA and RNA sequencing of select patient-derived organoid models (SMBO-106, SMBO-109, SMBO114, SMBO-170, and SCBO-8)[35]. Among the 5 PDOs profiled, we observed whole genome duplication (WGD) in a subset of cells in SMBO-106 (Clone A), SMBO-109 (Clone E), SMBO-114 (Clone B) and SCBO-8 (Clone D), as demonstrated in the heatmap and clone-specific copy number profiles (Fig. 3A, B, and Supplementary Fig. 5A-C). In SMBO-

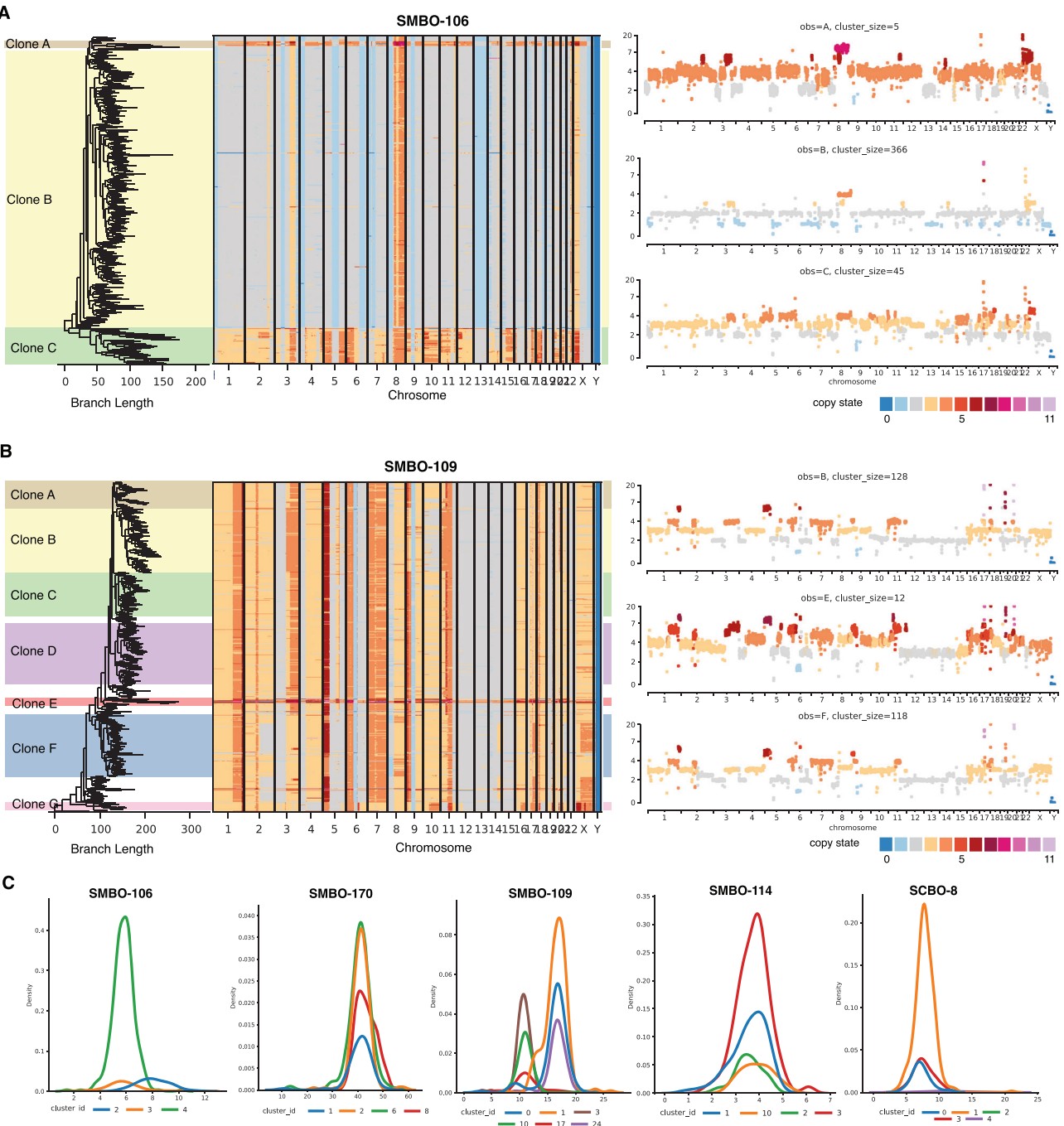

**Fig. 3 | Heterogeneity of *ERBB2* copy number as characterized by single-cell DNA sequencing. A**, **B** MEDICC2 analysis of inferred copy number phylogenies from single-cell DNA sequencing of SMBO-106 (**A**) and SMBO-109 (**B**) organoids. The branch length (left) represents an estimate of the number of genomic alterations. The heatmap (middle) represents copy number state profiles for individual organoid cells. Right: Total copy number across the genome for selected phylogenic clones. **C** Density distribution of *ERBB2* copy number of clusters of 25 or more cells identified by k-means clustering.

106, we also observed a population of cells (Clone C) that underwent WGD followed by widespread chromosomal arm level losses. In contrast, SMBO-170 had a largely homogeneous population of tumor cells with an estimated 42 copies of *ERBB2*, although a subset of cells demonstrated shallow deletion of *TP53* (Supplementary Fig. 5D). Among the 5 PDO models analyzed, SMBO-106 and SMBO-109 had heterogenous *ERBB2* amplification, whereas SMBO-170, SMBO-114 and SCBO-8 had largely homogeneous *ERBB2* copy number profiles (Fig. 3C). SMBO-109 was again notable for a heterogeneous population of *ERBB2* amplified cells with ~39% of tumor cells having 11 copies of the *ERBB2* gene and 55% 17 copies. Phylogenic analyses revealed that these

two populations likely diverged early during clonal evolution. Conversely, the SMBO-109 organoid line had clonal mutations in both *TP53* and *RB1* (both with 100% cancer cell fraction (ccf)) in addition to homogenous copy number at the respective gene loci suggesting that these alterations arose early during disease pathogenesis.

To better understand the heterogeneity and transcriptional features of individual urothelial cancer cells, we also analyzed the same 5 PDO models using single-cell RNA sequencing. Based on the consensus classification schema, three of the PDO models had luminal gene expression signatures (SMBO-170, SCBO-8, and SMBO-114), whereas the other two basal gene expression signatures (SMBO-106, SMBO-109)

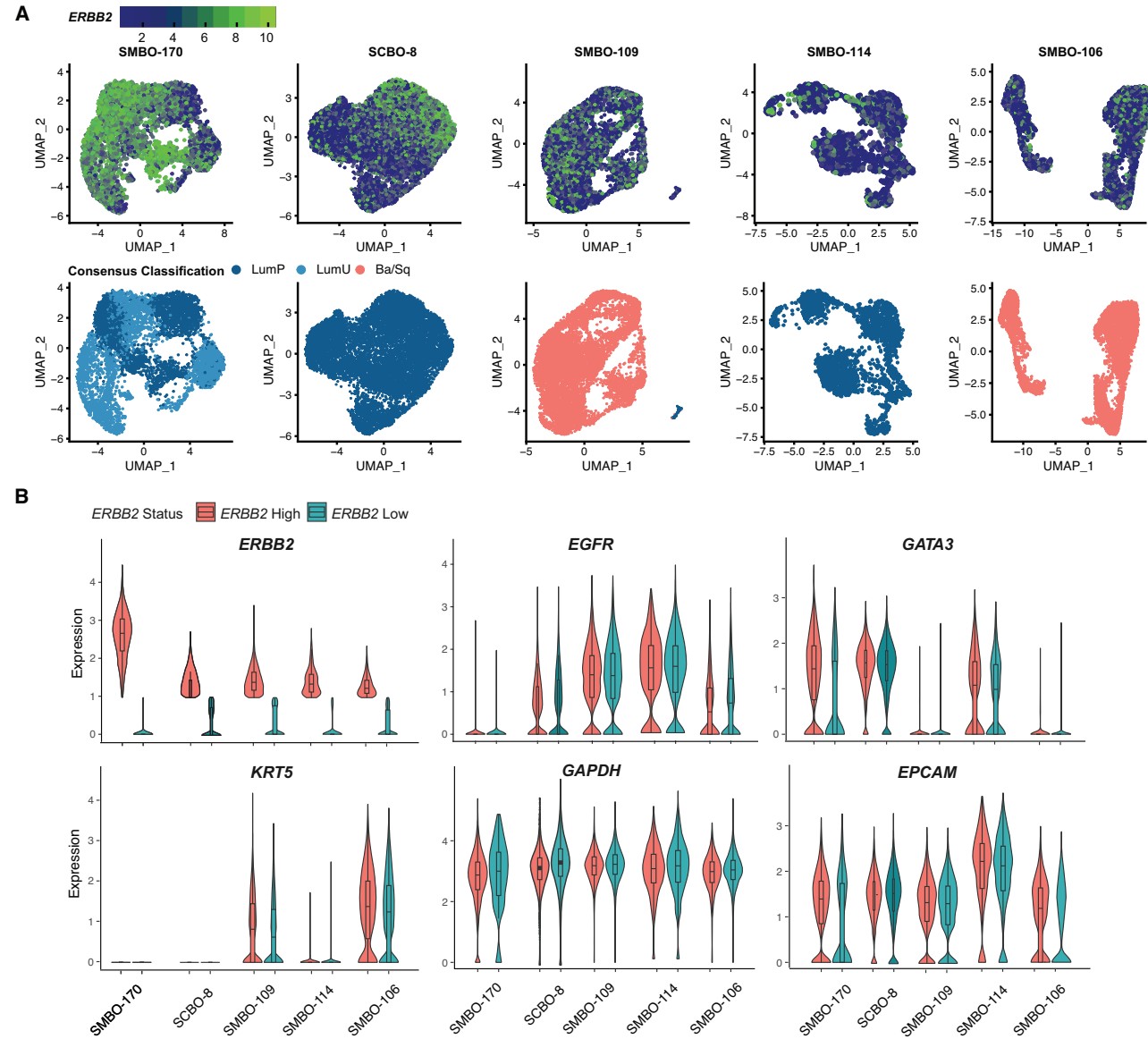

**Fig. 4 | Single-cell RNA sequencing analysis of urothelial cancer patient derived organoid models. A** *ERBB2* expression (top) and transcriptional subtype of individual tumor cells based on the urothelial consensus transcriptional classification schema (bottom). **B** Violin plots demonstrating expression of *ERBB2* and other select genes in cells with high and low *ERBB2* expression levels. Detailed information for the box plot definitions is provided in Source Data.

(Fig. 4A). Consistent with the homogeneity of *ERBB2* amplification noted by scWGS and scRNAseq in the SMBO-170 organoids, this model had high and largely uniform overexpression of *ERBB2* transcripts. In contrast, SCBO-8, SMBO-109, SMBO-114 and SMBO-106 had heterogenous populations of high and low *ERBB2* expressing tumor cells. Of note, all 5 models also contained a sub-population of cells co-expressing high levels of *ERBB2* and *EGFR*, with co-expression of *ERBB2* and *EGFR* particularly common in SMBO-109 (Fig. 4B, and Supplementary Fig. S6). This pattern of *ERBB2* and *EGFR* co-expression in a subset of cancer cells may be of clinical significance as elevated *EGFR* expression can promote T-DXd resistance by suppressing T-DXd internalization[32]. In sum, our single-cell data suggests heterogeneity of *ERBB2* copy number and gene expression in urothelial cancer organoid lines.

## Determinants of trastuzumab deruxtecan (T-DXd) sensitivity in urothelial cancer

Several HER2-selective kinase inhibitors, including lapatinib, tucatinib, and neratinib, have clinical efficacy in patients with *ERBB2* amplified and mutant breast cancer[36], whereas disappointing results have been reported with HER2 kinase inhibitors in patients with urothelial cancer[16]. To explore the mechanistic basis for these disparate clinical results, we leveraged our biobank of patient-derived organoids to compare the dependence of downstream signaling pathways, including ERK and AKT, on HER2 activation. As shown in Fig. 5A, treatment of BT-474 (*ERBB2* amplified) breast cancer cells with increasing concentrations of neratinib (up to 250 nM) for 1 h led to a dose dependent decrease in the expression of downstream effectors of HER2 including phosphorylated AKT (pAKT) and ERK (pERK). Notably, the concentration of neratinib required to downregulate pERK expression was significantly higher for SMBO-170, SCBO-8, SMBO-109, SMBO-114, and SMBO-106 urothelial cancer organoids than BT-474 breast cancer cells. Similarly, higher concentrations of neratinib were required to downregulate pAKT expression in SMBO-170 versus BT-474, and neratinib had minimal to no effect on pAKT expression in SCBO-8, SMBO-109, SMBO-114, and SMBO-

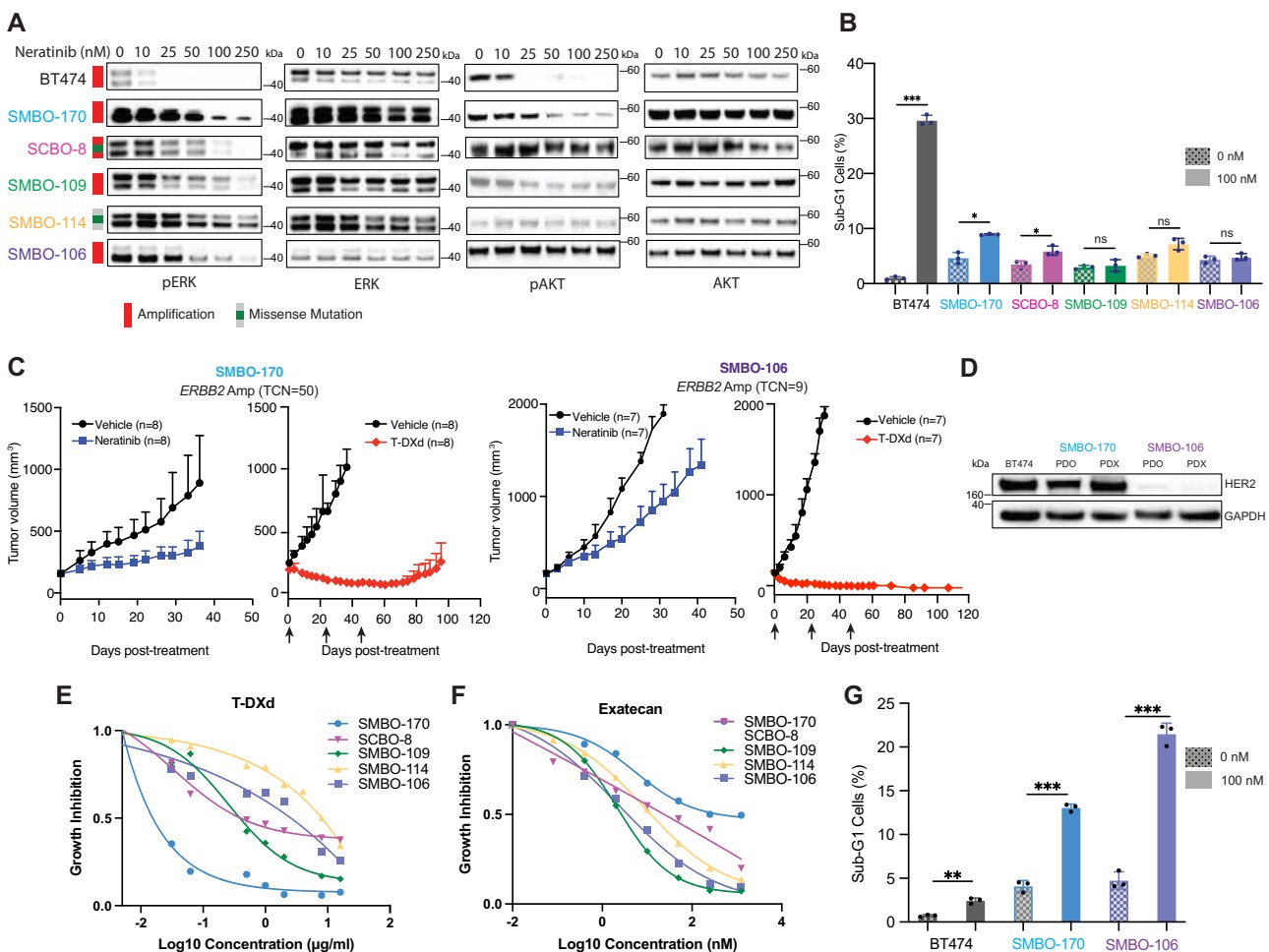

**Fig. 5 | HER2 dependence is not required for trastuzumab deruxtecan sensitivity in urothelial cancer. A** Urothelial cancer organoids (SMBO-170, SCBO-8, SMBO-109, SMBO-114, and SMBO-106) and BT-474 (*ERBB2* amplified breast cancer) cells were treated with neratinib at the concentrations indicated for 1 h. Changes in the expression of pERK (Thr202/Tyr204), ERK, pAKT (ser473) and AKT as a function of neratinib concentration were determined by immunoblot. **B** Cell death quantitated by flow cytometry following 48 h of treatment with neratinib at a concentration of 100 nM. BT474: *p* < 0.001; SMBO-170: *p* = 0.0166; SCBO-8: *p* = 0.0177; SMBO-109: *p* = 0.6674; SMBO-114: *p* = 0.0626; SMBO-106: *p* = 0.3698. **C** Mice with established SMBO-170 or SMBO-106 xenografts were treated with neratinib (20 mg/kg; p.o. QD × 5) or T-DXd (10 mg/kg, i.v., once every 3 weeks for 9 weeks). TCN, total copy number. SMBO-170: *p* < 0.001 vehicle vs neratinib; *p* < 0.001 vehicle vs T-DXd; SMBO-106: *p* < 0.001 vehicle vs neratinib; p < 0.001 vehicle vs T-DXd. Two-way ANOVA test (Prism) was used for statistical analysis without adjustment. Data are presented as mean values ± SD. Only the upper error bars are displayed for clarity. **D** Immunoblots comparing expression of HER2 in protein lysates extracted from BT474 breast cancer cells and SMBO-170 and SMBO-106 urothelial cancer organoids and xenografts. **E** Cell viability as determined by MTT assays for urothelial organoids treated with T-DXd for 8 days with concentrations ranging from 0 - 16 µg/ml. **F** Cell viability as determined by MTT assay for organoids treated with the DNA topoisomerase 1 inhibitor exatecan (DX8951f), an analog of the T-DXd payload deruxtecan, for 3 days with concentrations ranging from 0 - 1250 nM. **G** Cell death quantitated by flow cytometry after 48 h of exatecan treatment (100 nM). BT474: *p* = 0.005; SMBO-170: *p* < 0.001; SMBO-106: *p* < 0.001. Data are representative of three independent experiments for (**A** and **D**). Results are shown as means ± SD for three experimental replicates for (**B** and **G**). *P* value were determined by two-sided unpaired t test with Welch's correction (Prism) for (**B** and **G**). An example of the gating strategy for B and G is provided in Figure S9 in Supplementary Information file. Significance denoted as **p* < 0.05, ***p* < 0.01, ****p* < 0.001. ns, not significant. Source data for (**A**–**G**) are provided in Source Data.

106 organoids. These data suggest that AKT activation is not dependent on HER2 kinase activation in most urothelial cancer cells, including at least a subset of those with *ERBB2* mutation, which contrasts with prior studies suggesting that breast cancers with HER2 amplification are selectively dependent on AKT signaling[36]. Notably, we observed similar effects of neratinib on pERK and pAKT expression irrespective of whether SMBO-106 and SMBO-170 cells were grown as organoids or 2D cell lines (Supplementary Fig. 7A). Additionally, treatment with neratinib induced significantly more cell death of BT-474 breast cancer cells than urothelial cancer organoids as determined by flow cytometry analysis (Fig. 5B, Supplementary Fig. 7B).

We next directly compared the efficacy of T-DXd to neratinib in mice with established SMBO-170 (HER2 3+ expression) and SMBO-106 (HER2 1+ expression) xenografts. In both models, neratinib only slowed the growth of established xenografts [70.8% tumor growth inhibition (TGI) of SMBO-170 and 45.8% TGI of SMBO-106 at day 41, (Fig. 5C, and Supplementary Fig. 7C)]. T-DXd was significantly more effective in both patient derived models inducing tumor regression in SMBO-170 (maximal regression of a mean 77.2% at day 36) and complete responses in all mice bearing SMBO-106 xenografts. Durable complete responses to T-DXd were also observed in all mice with even larger (mean 500 mm³) SMBO-106 xenografts pretreatment (*n* = 8, Supplementary Fig. 7D). Additionally, T-DXd had more durable antitumor activity in both the SMBO-170 and SMBO-106 xenograft models than enfortumab vedotin, a Nectin-4-targeted ADC that has recently emerged as a new standard of care for the treatment of patients with advanced urothelial cancer (Supplementary Fig. 8A, B)[37]. In sum, these

data suggest that oncogenic dependence of urothelial cancer cells on HER2 is not required for T-DXd sensitivity. The results also support the broader clinical testing of T-DXd in patients with urothelial cancers with low HER2 expression (Fig. 5D).

To clarify the basis for the extreme sensitivity of SMBO-106 PDXs to T-DXd, we next compared the in vitro sensitivity of SMBO-170, SMBO-106, SMBO-109 and SMBO-114 organoids to T-DXd and exatecan, a topoisomerase I inhibitor and deruxtecan analogue. As shown in Fig. 5E, F, whereas SMBO-170 organoids were more sensitive to T-DXd than SMBO-106 organoids, as defined by MTT viability assays, SMBO-106 organoids were more sensitive to exatecan. Furthermore, treatment with exatecan induced significantly more cell death of SMBO-106 versus SMBO-170 cells (Fig. 5G, Supplementary Fig. 7E). In a broader panel of 15 urothelial cancer PDO models, SMBO-170 was notable for its relative resistance to exatecan (Supplementary Fig. 8C). Together with the greater T-DXd sensitivity of SMBO-106 versus SMBO-170 xenografts, the data suggest that induction of cell death by the cytotoxic payload, may be a more important predictor of response to HER2-targeted ADCs in a bladder cancer context than HER2 expression or HER2-oncogene dependence.

The above preclinical data suggesting greater anti-tumor activity with HER2 ADCs than kinase inhibitors in urothelial cancer is supported by our real-world clinical experience with T-DXd in patients with metastatic urothelial cancer (Fig. 6A). While the median progression free survival of patients with urothelial cancer treated with T-DXd was only 4.95 months (95% CI, 2.47-11 months), complete and durable responses were observed with 8 patients remaining on drug for 12 months or longer (Fig. 6B, C). Of particular note, 3 of 4 patients with the longest duration of response to T-DXd (41+, 31 and 15+ months) had both ERBB2 amplification and mutation suggesting that concurrent amplification and mutation may be a predictor of exceptional response to HER2-targeted ADCs in patients with urothelial cancer. Additionally, responses to T-DXd were observed in patients with urothelial cancers that harbored co-mutations in TP53 (73%) and RB1 (33%), genomic alterations that have been associated with resistance to targeted therapies in other clinical contexts[38,39].

Notably, consistent with the frequent ERBB2 mutational discordance between primary and metastatic tumor samples, three patients in our real-world cohort had discordance of ERBB2 mutational status between cfDNA and archival tumor tissue (Fig. 6A). Additionally, a patient with primary resistance to T-DXd was found to have HER2 3+ expression in the primary tumor but no HER2 expression in a metastatic site collected at the time of disease progression on T-DXd (Fig. 6D). This case in combination with the frequent discordance of ERBB2 status between primary and metastatic disease sites and tumor and cfDNA profiling suggests that archival tumor tissue should be used with caution for determining ERBB2 mutational and HER2 protein expression status in patients with metastatic urothelial cancer being considered for treatment with HER2-targeted ADCs and that HER2 mutational and expression heterogeneity is common in patients with urothelial cancer and may be a clinically relevant mechanism of T-DXd resistance.

## Discussion

Despite the high prevalence of ERBB2 mutation and amplification in bladder and upper tract cancers, HER2 directed therapies have only recently demonstrated sufficient antitumor activity to warrant their routine use in patients with advanced urothelial cancers. Most notably, the HER2-targeted antibody drug conjugate trastuzumab deruxtecan (T-DXd) received a tumor agnostic FDA accelerated approval in 2024 for the treatment of patients with advanced solid tumors with HER2 3+ expression by immunohistochemistry, including urothelial cancers[19]. Compelling clinical activity has also been reported with a second HER2-targeted ADC, disitamab vedotin, alone and in combination with anti-PD1-targeted immunotherapy in patients with urothelial

cancer[40,41]. Assessment of HER2 expression and consideration of treatment with HER2-targeted ADCs has thus emerged as a component of the standard management of patients with treatment-refractory advanced urothelial cancers.

To elucidate the mechanistic-basis for the lack of clinical activity of older HER2-targeted therapies in patients with urothelial cancer and to guide the clinical development of novel HER2-targeted therapies including HER2-targeted ADCs, we performed integrated molecular and clinical analyses of patients with urothelial cancer enrolled on a prospective molecular tumor profiling initiative. We also leveraged this prospective tumor sequencing initiative to develop and biologically characterize novel ERBB2 amplified and mutant urothelial cancer patient-derived organoid and xenograft models that better reflect the genomic and biologic complexity of urothelial cancers than older 2D cell line models. Our studies of urothelial cancer organoids derived from tumors classified clinically as ERBB2 amplified revealed that these models lacked the HER2 oncogenic dependence characteristic of HER2 amplified breast cancer cells, with higher concentrations of neratinib required to inhibit downstream effectors of HER2 signaling including ERK and AKT. Consistent with the clinical experience, ERBB2 mutant and amplified urothelial cancer patient-derived xenograft models were also significantly more sensitive to T-DXd than neratinib.

A notable finding of our study was the high degree of ERBB2 mutational discordance between patient-matched primary and metastatic tumor pairs and the urothelial cancer organoid models and the tumors from which they were derived. These findings suggest that ERBB2 amplification and mutation is often a later event in urothelial cancer pathogenesis and not present in all tumor cells in a significant minority of patients with urothelial cancer, which contrasts with breast cancer, a disease in which ERBB2 amplification has been shown to be an early initiating event and highly concordant between primary and metastatic disease sites[42,43]. One clinical implication of these findings is that molecular profiling of archival tumor samples collected from the primary disease site may not provide an accurate assessment of ERBB2 mutational and HER2 expression status in patients with urothelial cancer, in particular in patients with significant intervening therapy. The results also suggest similarities between urothelial and esophagogastric cancers, a cancer type in which ERBB2 heterogeneity and selection for ERBB2 wildtype subclones has been shown to be a common mechanisms of resistance to HER2-directed therapies[44–47]. To further characterize the molecular heterogeneity of the patient-derived models at the single-cell level, we also analyzed select early passage urothelial PDO models using a multiplatform approach which included single-cell DNA and RNA sequencing. Single-cell DNA sequencing revealed heterogeneity of ERBB2 copy number profiles in SMBO-109 organoids, an organoid model with discordance of ERBB2 amplification status between the PDO and tumor from which it was derived. Single-cell RNA sequencing also identified significant ERBB2 expression heterogeneity with variable expression of EGFR also observed. The later may be of clinical significance as elevated EGFR expression levels promote EGFR/HER2 heterodimer formation suppressing T-DXd internalization and efficacy[32]. The degree of ERBB2 expression heterogeneity in our organoid models also support the use of ADCs such as T-DXd with cleavable linkers which have the potential to overcome HER2 expression heterogeneity via the bystander effect[48,49].

A surprising result of clinical trials of T-DXd in breast and lung cancers has been the degree of clinical benefit observed in patients with low or undetectable HER2 expression as defined by immunohistochemistry (IHC)[18,50]. While the reported objective response rates for T-DXd have been consistently higher in patients with HER2 3+ expression than in patients with lower HER2 expression levels across solid tumor types[19], T-DXd has clinically meaningful anti-tumor activity in patients with HER2-low and even HER2-ultralow breast and lung cancers[18,20,50]. Specifically, the DESTINY-Breast06 trial reported

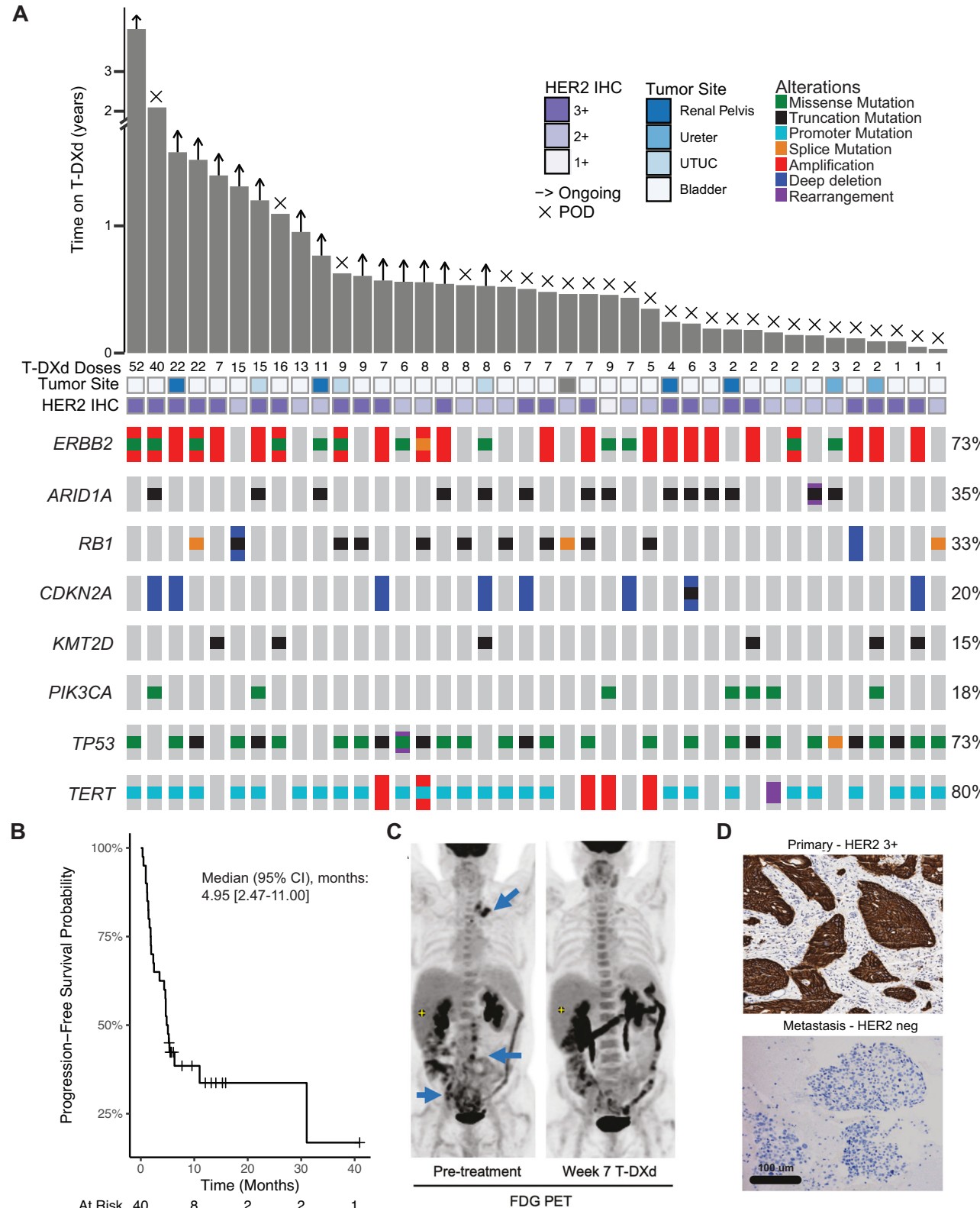

**Fig. 6 | Real-world clinical experience of patients with metastatic urothelial cancer treated with trastuzumab deruxtecan (T-DXd). A** Swimmer plot and corresponding OncoPrint of 40 patients with metastatic urothelial carcinoma treated with T-DXd. HER2 immunohistochemistry (IHC) score, tumor primary site and the number of T-DXd doses administered are indicated for each patient. **B** Kaplan-Meier curve of progression free survival in 40 patients with metastatic urothelial carcinoma treated with T-DXd. **C** Pre- and post-treatment (7 weeks) FDG-PET images of a urothelial cancer patient with HER2 2+ expression by IHC treated

with T-DXd who achieved a complete metabolic response with resolution of FDG uptake in supraclavicular, retroperitoneal, and pelvic lymph node metastases (blue arrows). **D** Pre-treatment and progression tumor biopsies collected from a patient with urothelial cancer with primary resistance to T-DXd demonstrating HER2 3+ expression in the pretreatment tumor, but no HER2 expression in a metastasis collected following disease progression. Representative images shown in 6D are from one single experiment and the scale bar is applicable for both images. Source data for (**A**, **B**) are provided in Source Data.

significantly longer progression-free survival with T-DXd (13.2 months) versus chemotherapy (8.1 months) in patients with HER2-low (IHC 1+ or IHC 2+/ISH-) or HER2-ultralow (IHC 0 with membrane staining) breast cancer resulting in its FDA approval for this molecularly defined patient population[51]. Additionally, the DESTINY-PanTumor01 basket study, which enrolled HER2 mutant solid tumors with low HER2 expression, reported a 29.4% response rate, including objective responses to T-DXd in 2 of 7 patients with urothelial cancer[52]. In the case of lung cancers with *ERBB2* mutation, which typically express low levels of HER2 protein[50], *ERBB2* mutation has been shown to enhance ADC internalization[53]. This enhanced internalization of the HER2/T-DXd complex in the context of HER2 mutation may provide a biologic basis for the exceptional responses observed in our real-world cohort in patients with *ERBB2* co-amplification and mutation as such patients would be predicted to have both high levels of ADC target expression and enhanced ADC internalization as compared to those with wildtype HER2.

Our real-world clinical cohort had several limitations that will need to be addressed as part of future preclinical and clinical studies. Our clinical experience with T-DXd was biased toward patients with higher levels of HER2 expression as the current tumor agnostic FDA-authorization is currently restricted to patients with HER2 3+ expression. The standard treatment approach to patients with locally advanced and metastatic urothelial cancer has also evolved significantly over the past several years with the introduction of immune checkpoint inhibitors and the Nectin-4-targeted ADC enfortumab vedotin and the recent demonstration of superiority of the enfortumab vedotin plus pembrolizumab combination versus cytotoxic chemotherapy[37]. Our data demonstrating dramatic and durable responses to T-DXd in patients with co-mutation and amplification of *ERBB2*, suggest that this may represent a molecularly defined population for which HER2-targeted ADCs may be superior to Nectin-4-targeted therapies. However, as our patient derived organoid/xenograft biobank was generated prior to the widespread use of enfortumab vedotin, we could not directly study the activity of T-DXd in models derived from patients with acquired enfortumab vedotin resistance. Future studies should thus seek to develop patient-derived models from tumors that exhibit intrinsic or acquired resistance to enfortumab vedotin, in particular, models derived from metastatic sites as Nectin-4 and HER2 have been shown to be differentially expressed in patient-matched primary and metastatic tumor sites[54,55]. Additionally, the availability of HER2-targeted ADCs with different cytotoxic payloads may allow for sequential therapy for patients with acquired resistance driven by resistance to the cytotoxic payload as opposed to loss of target expression or impaired ADC internalization or trafficking, a hypothesis that could be directly tested in patient-derived models.

The early clinical experience with T-DXd[56] and disitamab vedotin[41] also suggest that the latter may have greater activity when combined with pembrolizumab or other immune checkpoint inhibitors, possibly due to difference in the ability of the vedotin versus deruxtecan payload to induce immunogenic cell death[57–59]. There is thus an urgent need to development immune competent models of *ERBB2* amplified and mutant urothelial cancer to allow for direct comparison of HER2 ADCs with different payloads to identify those with the greatest potential to enhance the activity of immune checkpoint inhibitors. Such models will also be critical for the preclinical evaluation of novel immune-based approaches such as vaccines or cellular therapies for the treatment of *ERBB2* mutant and amplified urothelial cancers. Finally, the exquisite sensitivity of SMBO-106 (low HER2 expression) xenografts to T-DXd which mirrored the high degree of sensitivity of SMBO-106 organoids to exatecan-induced cell death suggests that the cytotoxic payload may be a more critical predictor of HER2-targeted ADCs response than HER2 expression levels, supporting ongoing efforts to develop functional precision oncology diagnostic platforms

capable of quantitating payload sensitivity pre-treatment thus allowing for greater individualization of therapy[60]

## Methods

All specimens were obtained from patients in accordance with institutional review board (IRB) approval at Memorial Sloan Kettering Cancer Center (MSKCC) or Columbia University Medical Center. Patient derived models were generated under MSKCC IRB approved protocols 06-107, 12-245, or 19-105 or Columbia University Medical Center AAAN8850 or AAAT2128. Procedures for animals were performed under approved MSKCC Institutional Animal Care and Use Committee (IACUC) protocols (13-07-006, 04-03-009).

### Patient cohort

We analyzed MSK-IMPACT (Memorial Sloan Kettering - Integrated Mutation Profiling of Actionable Cancer Targets) data for 56,801 unique tumor samples analyzed as part of an institution-wide prospective sequencing study (ClinicalTrials.gov identifier: NCT01775072) at Memorial Sloan Kettering Cancer Center (New York, NY)[61]. Samples with an estimated tumor purity of less than 20% and those with a tumor mutational burden exceeding the 95th percentile of their respective cancer type were excluded. The final cohort was comprised of 42,415 tumors, of which 2,035 were bladder and upper tract cancers. All urothelial cancer tumors were centrally reviewed (by H.A.A.) to confirm the diagnosis of urothelial cancer and to assess for the presence of variant histologic subtypes and to determine stage and grade as per the World Health Organization (WHO) classification system[62]. To analyze real-world clinical outcomes of patients with metastatic urothelial cancer treated with trastuzumab deruxtecan (T-DXd), we identified all patients with urothelial cancer treated with T-DXd at MSKCC prior to January 30, 2025. One patient was excluded from the analysis based on prior treatment with the HER2-targeted ADC disitamab vedotin within the context of an ongoing clinical trial.

### Establishment of patient derived organoid (PDO) and patient derived xenograft (PDX) models

For generation of patients derived models, patients were consented to MSKCC IRB 06-107, 12-245, or 19-105 or Columbia University Medical Center AAAN8850 or AAAT2128. After pathologic review, tumor specimens were transported on ice to the research laboratory in organoid culture media consisting of hepatocyte culture media supplemented with 10 ng/ml epidermal growth factor (EGF, Corning), 5% charcoal-stripped fetal bovine serum (CS-FBS, Gemini), 100 μg/ml Primocin (InvivoGen), 1x Glutamax (Gibco), and 10 μM ROCK inhibitor (Y-27632, STEMCELL Technologies).

To establish organoid models, tissue dissociation was performed using a modified protocol from that previously described[63]. In brief, tissue was minced with scissors and incubated in 10 mL of organoid culture media with a 1:10 dilution of collagenase/hyaluronidase (STEMCELL Technologies) at 37 °C for 15 min. The digestion was stopped with 10 ml of organoid culture media, and the content was filtered through a 100 mm strainer (Miltenyi Biotec), washed and then spun down at 300xG for 5 min. Dissociated cell clusters (~5000 cells/ml) were resuspended in 90% Matrigel (Corning)/organoid culture media and 50–100 μl was added per well in a pre-warmed Cellstar Cell Culture 24 well plate (Greiner Bio-One). Bubbles were solidified after 15–20 minutes of incubation at 37 °C and 5% $CO_2$, and 1 ml organoid culture media was added to each well. Media was changed every 2–4 days and organoids were passaged every 1-2 weeks. All PDO models were generated directly from patient tumors except for SMBO-114, previously identified as UCC14, which was generated from an established patient derived xenograft (PDX) as previously described[64].

All animal studies were performed under the auspices of a protocol approved by the MSKCC IACUC (13-07-006 and 04-03-009). To establish xenograft models, tumor implantation was performed as

previously described[64]. Briefly, specimens were cut into small pieces and then implanted subcutaneously into one or two flanks of 6-8 week-old male immunocompromised NOD-SCID *IL2Rg*[−/−] (NSG ®) mice (The Jackson Laboratory) via a 4 - 5 mm skin incision over the mid-lumbar spine. The exception was SMBO-170 PDX, which was established by subcutaneous injection of SMBO-170 PDOs. PDX models were serially transplanted for expansion when the tumor volume reached 0.5–1 cm³.

Mice were housed under controlled environmental conditions of $21 \pm 1.5\,°C$ temperature, $55 \pm 10\%$ humidity and a 12 h light–dark cycle (lights were on from 6:00 to 18:00). Mice treated with T-DXd, enfortumab vedotin (EV) or neratinib were monitored daily for clinical signs of distress (weight loss, hunched posture, diarrhea, dehydration, anemia, tumor ulcerations) and tumor volumes were recorded twice per week. Animals were humanely euthanized when tumors approached maximal allowed tumor size. In cases in which tumors were eradicated by the treatment, mice were monitored for potential tumor regrowth for up to 120 days after the last treatment. Maximal tumor size was defined as 10% body weight by volume and this maximal tumor size was not exceeded.

A summary of tumor characteristics for the 45 urothelial cancer patient-derived xenograft and organoid lines and the corresponding clinical data from the patients from which they were derived is provided in Supplementary Table 1.

### Immunohistochemistry (IHC), haematoxylin–eosin (H&E), and DNA fluorescent in situ hybridization (DNA-FISH) staining

Roughly 2 million cells digested from PDOs using TrypLE were loaded onto 15 ml tubes with agarose filling for embedding. After washing with PBS and fixing with 4% Paraformaldehyde (ThermoFisher), PDO pellets were sectioned at 5 µm and mounted onto glass slides.

IHC staining was performed using a Her2/neu antibody (Ventana #790-2991, 1:10 dilution) using the BOND RX (Leica Biosystems) system. HER2 protein expression was scored (by H.A.-A.) according to ASCO/CAP (2018) guidelines for HER2 analysis of invasive breast cancers[65]. H&E staining was performed following standard protocol and images were captured using in-house digital pathology platform.

FISH analysis was performed using a two-color *ERBB2*–centromeric chromosome 17 (CEP17) probe on sequential sections of frozen tissue blocks. Probe labeling, tissue processing, hybridization, post-hybridization washing, and fluorescence detection were performed based on MSKCC standard operating protocol for clinical testing. Slides were scanned using a Confocal microscope and assessment of *ERBB2* amplification was performed using ASCO/CAP (2018) guidelines[65]. Amplification was defined by an Her2/Cep17 ratio of ≥2.0 or ≥ 6 copies of the gene.

### Targeted sequencing

To compare the genomic profiles of PDX/O models from the tumors from which they were derived, FFPE tumors and matched normal tissue (typically blood), and cell pellets from PDOs were sequenced using the MSK-IMPACT assay. Detailed methodology regarding DNA extraction, library preparation, sequencing and bioinformatic analysis have been described previously[21,61]. Oncogenic and likely oncogenic alterations were annotated using OncoKB[66]. Tumor mutational burden (TMB) was calculated by dividing the number of sequence mutations identified by MSK-IMPACT by the total genomic area where mutations were reported, based on the version of the assay used. Copy number analysis was performed using FACETS, as previously described[67].

### Analysis of the TCGA cohort and bulk RNA-seq deconvolution analysis

RNA-seq raw sequencing data from the TCGA bladder cohort (BLCA) were downloaded from the NIH Genomic Data Commons (https://gdc. cancer.gov/). Sequencing reads were aligned against human genome assembly hg19 by STAR 2-pass alignment[68]. QC metrics, for example general sequencing statistics, gene feature and coverage, were then calculated based on the alignment result through RSeQC[69]. RNA-seq gene level count values were computed by using the R package GenomicAlignments[70] over aligned reads with UCSC KnownGene[71] in hg19 as the base gene model. The Union counting mode was used and only mapped paired reads after alignment quality filtering were considered. Gene level FPKM (Fragments Per Kilobase Million) and raw read count values were computed using the R package DESeq2[72] and then TPM (Transcripts Per Kilobase Million) values were derived from FPKM values. For bulk RNA-seq deconvolution we employed the ssGSEA[73] method. Signature gene lists of immune cell types as well as immune features were obtained according to Bindea et al[74]. and Senbabaoglu et al[75].

### Whole exome sequencing and analysis

Whole exome sequencing was performed from newly extracted DNA or through re-capture of existing MSK-IMPACT sequencing libraries using the xGen Exome Research Panel v2 (IDT), with libraries sequenced using a NovaSeq 6000 as 100 bp paired end mode to a target coverage of 150x (for tumor) and 70x (for normal). Whole exome sequencing data was analyzed using the TEMPO pipeline (Time-Efficient Mutational Profiling in Oncology; https://github.com/mskcc/tempo). Briefly, reads were aligned to the human genome (hg38) using BWA MEM, followed by post-processing using the Picard MarkDuplicates tool and GATK. Somatic variant calling was performed using MuTect2[76], Strelka2[77], Manta[78] and Delly[79]. Mutational signature analysis was performed as previously described[80]. Mutation phylogeny between primary and metastatic tumors was inferred using the union of somatic mutations called in paired samples and performed using the R package ape[81]. Mutational signatures were inferred from single-nucleotide mutations and the fraction of mutations attributable to each of 30 known mutational signatures[82] was determined using a basin-hopping algorithm (https://github.com/mskcc/mutation-signatures). Signatures with a known common source of somatic hypermutation were considered together (e.g., signatures 6, 14, 15, 20, 21 and 26 as mismatch-repair deficiency/MSI-associated).

### Single-cell RNA sequencing (scRNA-seq), and single-cell whole genome sequencing (DLP)

For single-cell RNA sequencing (scRNA-seq) and single cell whole genome DNA sequencing (DLP), PDOs were dissociated as described above and passed through a 30 mm filter to remove cell aggregates. The singlet cells were then resuspended in PBS containing 0.04% BSA, counted on a Countess II automated cell counter (Thermo Fisher), and then divided into two tubes, one each for scRNA-seq and DLP.

For scRNA-seq, -12,000 cells were loaded per lane on 10X Chromium microfluidic chips. Single-cell capture, barcoding, and library preparation were performed using the Chromium Single Cell 3' library (10x Genomics) and Gel bead Kit V3 according to the manufacturer's protocol (CG000204). The resulting DNA library was sequenced on a NovaSeq 6000 in a PE28/91 read run resulting in a median of -22,000 reads per cell with a median sequencing saturation of 27%, -1750 genes per cell on average. Cell ranger v3.1.0 (10X Genomics) was used to demultiplex and process the FASTQ files, align reads to the GRCh38 human transcriptome and generate a gene-cell counts matrix. Downstream analyses were performed with the Seurat package v4.0[83]. Cells with a high fraction of mitochondrial contents (>10%), low library complexity (cells expressing very few unique genes), and doublets (identified using DoubletFinder v2.0) were filtered. On average, we sequenced 9468 cells per sample with 21,700 median reads per cell and 1760 median genes per cell. After filtering the diploid cells and cells with high mitochondrial contents, we obtained on average 5528 high quality cells/sample for downstream analysis. Data normalization

and dimension reduction (RunUMAP) were performed using default parameters. PDOs were assigned to transcriptional subtypes as per the consensus classification using the consensusMIBC package (v. 1.1) in R[34].

For DLP, library construction was carried out as described previously[35]. In brief, cells at a concentration of 1 million cells/ml were fluorescently stained using a combination of CellTrace CFSE Cell Proliferation Kit (ThermoFisher) and LIVE/DEAD Fixable Red Dead Cell Stains (ThermoFisher), then centrifuged to remove excessive stain, followed by resuspension in PBS with 0.04% BSA. Single-cell suspensions were loaded into a piezoelectric dispenser (cellenONE) and spotted into open nanowell arrays (SmartChip, TakaraBio) with unique dual index barcodes. After confirming the occupancy and cell state by fluorescent imaging, the chip was processed through a series of reagent addition, heat lysis, tagmentation, neutralization, recovery, and purification steps. Pooled single-cell libraries were analyzed using the Aglient Bioanalyzer 2100 HS kit and then sequenced on a NovaSeq 6000 (PE150 reads).

DLP data was demultiplexed into cell specific FASTQ files and processed using a custom pipeline (https://github.com/shahcompbio/single_cell_pipeline). Copy number phylogenies were inferred from allele-specific copy number data using MEDICC2 based on the minimum number of evolutionary events (including LOH, WGD, and segmental gains and losses of arbitrary size) needed to transform one genome into another[84].

## Lysate preparation and western blot analysis

To prepare whole cell lysates for western blot analysis, organoids were collected around similar confluency with Matrigel and were centrifuged at $21,000 \times g$ for 10 minutes to remove Matrigel from cell pellets as much as possible. Organoid cell pellets were lysed in whole cell lysis buffer (150 mM NaCl, 10% glycerol, 1% IGEPAL, 0.5% sodium deoxycholate, 2 mM EDTA, 1 mM NaF, 1 mM Na3VO4, 1 mM Na2MoO4, 0.1% SDS, 20 mM Tris-HCl, pH 8) with proteinase and phosphatase inhibitors (Fisher Scientific). After sonication for 10 min, total protein lysates (20 ug, quantified by Bradford assay) were loaded on SDS-PAGE gels (NuPAGE 4–12% Bis-Tris Protein Gels, Invitrogen). Membranes were probed using the following primary antibodies from Cell Signaling Technology: HER2 (#4290), EGFR (Y1173) (#2232), HER3 (#12708), ERK (#9102), p-ERK (T202/Y204) (#9101), AKT (#9272), p-AKT (Ser473) (#9271), Rb (4H1) (#9309), GAPDH (#2118), GATA3 (#5852), CD44 (#37259), and KRT14 (#30731), or from Abcam: KRT5 (#52635), and Nectin-4 (#155692). Primary antibodies were used at 1:1000 TBST/3% BSA with incubation overnight at 4 °C. Secondary antibodies were used at 1:1000 TBST for 1 h at room temperature. Proteins were visualized using Amersham ImageQuant 800.

## Drug sensitivity assays

For cell cycle analysis, selected PDOs were converted into 2D cell lines by seeding into regular T75 culture flask. Cells were replated at 300k/well into 6-well plates. After 24 h to allow adherence, cells were treated with 0 or 100 nM neratinib (MedChem Express), or 0 or 100 nM exatecan (MedChem Express) with three replicates for 2 days. The cells were harvested using 0.25% trypsin, 0.1% EDTA, and resuspended in 10% FBS medium with the collected culture media, washed with ice cold 1% bovine serum albumin PBS (BSA-PBS), fixed with cold (-20C) 1 ml 70% ethanol, stained with Propidium Iodide and then analyzed using a BD FACSAria-3 Cell Sorter. Cell populations were calculated after appropriate gating using FlowJo (10.8.1). The Priopidium Iodide intensity histogram was generated, and cells were gated into sub-G1, G0/G1, S and G2/M phase.

T-DXd and exatecan growth inhibition assays: PDOs were seeded at 30k/well in 5% Matrigel in Costar 96-well ultra-low attachment plates. After allowing 24 h for adherence, cells were treated with

0-16 µg/ml T-DXd for 8 days and 0-1250 nM exatecan for 3 days. Cell viability was determined by MTT assay.

Neratinib sensitivity in vitro: for PDOs, 500k cells/well were plated in Matrigel into 24-well ultra-low attachment plates at least 24 h before treatment. For BT474 cells (ATCC, HTB-20), 300k cells/well were plated into Corning Costar 6-well plates at least 24 h before treatment. For western blot analysis, cells were treated with 0-250 nM neratinib for 1 h.

For xenograft studies, tumor bearing mice ($n = 7–8$/group) were randomized to vehicle control or treatment arms when average tumor volume reached $100–500 \text{ mm}^3$ as specified in the figure legends. Neratinib (Selleck Chemicals, HKI-272) was administered by oral gavage (20 mg/kg; p.o. QD×5) with treatment continued until animals in the control group required sacrifice. Trastuzumab deruxtecan (T-DXd, Enhertu, Daiichi Sankyo) was administered intravenously (10 mg/kg, once every 3 weeks for 9 weeks). Enfortumab Vedotin (EV, Padcev, Pfizer and Astellas) was administered intravenously (5 mg/kg, on day 8, 11, 14). For SMBO-106, the neratinib and T-DXd studies were repeated with larger tumors with the treatment starting at a tumor volume of ~500 mm$^3$. Tumor growth inhibition (TGI, %) was calculated based on the ratio of average tumor volume of the treated group versus the control group.

## Statistics & reproducibility

No statistical method was used to predetermine sample size. For the MSK-IMPACT cohort, tumors were excluded from analyses if they had an estimated tumor purity of less than 20% or a tumor mutational burden exceeding the 95th percentile of their respective cancer type resulting in a final cohort of 42,415 tumors. Sample sizes were determined based on previous publications where at least 7-8 animals per group were analyzed for in vivo drug testing and 3 replicates per group were analyzed for in vitro drug testing. No data were excluded from the in vivo and in vitro analyses. The investigators were blinded to group allocation during data collection and analysis process.

Comparisons of categorical and continuous variables were performed using Pearson's chi-squared test and Mann-Whitney Wilcoxon test respectively and adjusted for multiple testing using the false discovery rate (FDR), whenever appropriate. All tests were two-sided. Unless otherwise stated, all statistical analyses were performed using r/Bioconductor (https://www.bioconductor.org/). Results were presented as mean ± standard deviation (SD).

## Reporting summary

Further information on research design is available in the Nature Portfolio Reporting Summary linked to this article.

## Data availability

Clinical MSK-IMPACT results have been contributed to AACR Project GENIE and are accessible via the AACR GENIE instance of cBioPortal. To access this data, users of the AACR GENIE website must agree not to try to identify patients included in the dataset or attempt to contact individual participants. All data in AACR GENIE is de-identified using the HIPAA Safe Harbor Method. We have also created a publicly available custom cBioPortal instance which includes the somatic mutational data for the 45 PDX/PDO models and the tumors from which they were derived: https://www.cbioportal.org/study/summary?id=blca_pdx_msk_2025. WES data has been deposited in dbGAP (Accession #phs001783.v8.p1: Exome Recapture and Sequencing of Prospectively Characterized Clinical Specimens From Cancer Patients): https://www.ncbi.nlm.nih.gov/projects/gap/cgi-bin/study.cgi?study_id=phs001783.v2.p1. dbGAP was chosen as the repository for whole exome sequencing data to comply with Institutional IRB and NIH requirements designed to ensure patient privacy. As per dbGAP policy, access to the data is restricted to qualified researchers for non-commercial, academic purposes. Public access to the scRNA-seq data

for the profiled patient derived organoid models has been deposited in the cellxgene database :(https://cellxgene.cziscience.com/collections/8a05eaf6-5680-41f2-9be7-eddc383b178a). scDNA-seq data for the same 5 patient derived organoid models has been deposited in the Zenodo database: (https://zenodo.org/records/17604401). Researchers interested in obtaining access to this dataset may submit a request to David Solit, M.D. (solitd@mskcc.org). Access will be granted to investigators for any academic use of this data. We aim to acknowledge requests within one week, and data will typically be made available within one week after access is granted. Data will remain accessible to approved investigators for the duration of the study's data-sharing period. Source data are provided with this paper.

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

## Acknowledgements
D.B.S acknowledges funding from the Mark Foundation, Cycle for Survival, NIH (R01-CA288692, P50-CA221745, and the Cancer Center Support Grant P30-CA008748). M.M.S acknowledges funding from NIH (R01-CA282785) and Mark Foundation (EDV24-0000000003). E.P acknowledges funding from NIH (R37CA276946). The authors thank members of the Kravis Center for Molecular Oncology and Diagnostic Molecular Pathology at Memorial Sloan Kettering Cancer Center for establishing and maintaining the MSK-IMPACT dataset; Elisa de Stanchina and members of the MSKCC Antitumor Assessment Core Facility, MSKCC High-Performance Computing Group, the MSKCC Integrated Genomics Operation (IGO), and the MSKCC Flow Cytometry, and Molecular Cytogenetic Core Facilities. We also thank all the patients who consented to donate their tumor tissues for this study, as well as the surgical teams who facilitated this work.

## Author contributions
Z.C., X.T., K.K. and D.B.S. designed the study. K.K., J.T., K.N., N.R., X.T., J.R.C. and M.M.S. generated the PDOs and PDXs. X.T., S.G., H.J., J.L., and E.S. designed and collected experimental data. Z.C., J.E.E., A.M., F.K., I.O., M.F.B., S.P.S. and N.M. performed statistical and bioinformatic analysis. M.B. and H.A.-A. analyzed tumor samples to characterize histologic subtypes. E.P., J.A.C., D.H.A., J.E.R., S.C., H.A.-A., G.I., and D.B.S. collected clinical samples and data. Z.C., X.T., J.E.E., F.K., E.P., M.F.B., E.S., S.C., M.M.S., H.A.-A., G.I. K.K., D.B.S. contributed to writing, review & editing. Supervision, D.B.S.

## Competing interests
I.O. reports professional services and activities from Ataraxis. E.P. has consulted with Merck, Janssen, Urogen Pharma, has served on Data Safety Monitoring Committee of CG Oncology, QED Therapeutics and received research funding from Janssen and Merck. M.F.B.: Consulting: AstraZeneca, Paige.AI. Intellectual property rights: SOPHiA Genetics. S.P.S. reports research funding from AstraZeneca and Bristol Myers Squibb. D.H.A. reports research funding from Seattle Genetics and Astellas. J.E.R. reports consultant/ advisory board fees for Astellas, AstraZeneca, Aktis, Bayer, BMS, Boehringer Ingelheim, Generate Biomedicines, Gilead, Loxo at Lilly, Merck, Samsung Bioepis, Seagen/Pfizer, Tyra Biosciences; grant/research support for Roche/Genentech; AstraZeneca, Seagen/Pfizer, Astellas, Acrivon, Loxo at Lilly, Bayer; and honorarium for Pfizer. Prior relationships (not active) include consultant role for Aadi Bioscience, EMD Serono, Hengrui, IMVAX, Janssen Oncology, Roche/ Genentech; and honorarium for EMD-Serono. S.C. has received institutional grant/funding from Daiichi-Sankyo and AstraZeneca, Share options in Totus Medicines, and consultation/Ad board/Honoraria from Daiichi-Sankyo, AstraZeneca, Lilly, Casdin Capital, and Pathos AI. H.A.A has consulted with AstraZeneca, Paige.AI, Pfizer and Janssen. G.I. has consulted Janssen, Mirati Therapeutics, Flare Therapeutics, Loxo/Lilly, and Bicycle Therapeutics. D.B.S. has served as a consultant for/received honorarium from Pfizer, Loxo/Lilly Oncology, Vividion Therapeutics, Scorpion Therapeutics, Elsie Biotechnologies, Inc., Fore Therapeutics, Function Oncology, Fog Pharma, Corramedical, Inc, Meliora Therapeutics, Inc., and BridgeBio. The remaining authors declare no competing interests.

## Additional information

¹Human Oncology and Pathogenesis Program, Memorial Sloan Kettering Cancer Center, New York, NY, USA. ²Weill Cornell Medicine, Graduate School of Medical Sciences, New York, NY, USA. ³Computational Oncology, Department of Epidemiology and Biostatistics, Memorial Sloan Kettering Cancer Center, New York, NY, USA. ⁴Halvorsen Center for Computational Oncology, Memorial Sloan Kettering Cancer Center, New York, NY, USA. ⁵Department of Surgery, Memorial Sloan Kettering Cancer Center, New York, NY, USA. ⁶Department of Medicine, Genetics and Development, Urology, and Systems Biology, Herbert Irving Comprehensive Cancer Center, Columbia University Vagelos College of Physicians and Surgeons, New York, NY, USA. ⁷Biostatistics, Department of Epidemiology-Biostatistics, Memorial Sloan Kettering Cancer Center, New York, NY, USA. ⁸Department of Pathology and Laboratory Medicine, Memorial Sloan Kettering Cancer Center, New York, NY, USA. ⁹Urology Service, Department of Urology, Memorial Sloan Kettering Cancer Center, New York, NY, USA. ¹⁰Marie-Josée and Henry R. Kravis Center for Molecular Oncology, Memorial Sloan Kettering Cancer Center, New York, NY, USA. ¹¹Antitumor Assessment Core Facility, Memorial Sloan Kettering Cancer Center, New York, NY, USA. ¹²Integrated Genomics Operation, Memorial Sloan Kettering Cancer Center, New York, NY, USA. ¹³Genitourinary Oncology Service, Department of Medicine, Memorial Sloan Kettering Cancer Center, New York, NY, USA. ¹⁴These authors contributed equally: Ziyu Chen, Xinran Tang. ✉e-mail: solitd@mskcc.org

