## [Transparent Peer Review file · Nature Communications]

Determinants of Sensitivity to HER2-targeted Antibody Drug Conjugates in Urothelial Cancer

Corresponding Author: Dr David Solit

Version 0:

Reviewer comments:

Reviewer #1

(Remarks to the Author)

1. Can authors mention differential HER2 alterations based on upper tract vs bladder primary tumors?
2. Is there an association between ERBB2 mutations and ERBB2 mRNA/protein expression?
3. The mechanism for sensitivity of HER2 mutations in particular to HER2 binding ADCs warrants some discussion. This appears similar to the phenomenon seen in NSCLC with HER2 mutations where greater internalization of the ADC appears to occur?
4. Another determinant of ADC activity is the linker, which has not been explored in this paper. This point could be discussed.
5. While T-DXd monotherapy has been approved after robust activity seen across tumors, the combination of T-DXd and nivolumab was underwhelming in a 2nd line trial (Galsky et al)- please incorporate into discussion with potential reasons- or is this possibly a powering issue and sample size in these trials. In contrast disitamab vedotin + PD1 inhibition appears highly active, likely due to a tubulin toxin payload, which may be more optimal? Can the molecular data in this study be used to offer reasons why a tubulin toxin may be more optimal?
6. The potential reasons for lack of sensitivity of ERBB2 mutated tumors to TKIs could be explored in this dataset given comprehensive profiling data. Is this possible?
7. One interesting experiment may have been to explore activity of T-DXd following resistance to EV or even exploring the combination. These points could be discussed?

Reviewer #2

(Remarks to the Author)

This manuscript investigates the landscape and significance of ERBB2 alterations in urothelial cancers using a large cohort of 2,035 patients with tumor genomic profiling data. The authors develop and biologically characterize novel patient-derived bladder cancer models with HER2 alterations, and assess their in vivo sensitivity to trastuzumab deruxtecan (T-DXd). These preclinical findings are complemented by data from a real-world clinical cohort of 28 patients with metastatic urothelial cancer treated with T-DXd. This is a timely and relevant study that provides valuable insights into HER2 biology and therapeutic response to HER2-targeted drugs in urothelial cancer.

We offer the following comments and suggestions to help improve the quality of the manuscript:

MAJOR COMMENTS:

Introduction and Discussion:

1. Given that the manuscript addresses both HER2 amplification and mutations, we recommend broader discussion of the clinical data on HER2-mutant tumors treated with T-DXd. In particular, consider integrating findings from DESTINY-PanTumor01 (Li et al., Lancet Oncology, 2024), which includes 7 patients with HER2-mutant urothelial carcinoma.
2. Although the approval and role of trastuzumab deruxtecan is widely covered, very little details are provided on disitamab vedotin, which has also shown impressive activity in HER2-altered urothelial cancer and is currently approved in China for this indication. We'd recommend a better description and discussion of this agent in the context of other anti-HER2 agents for urothelial cancer.

Results:

3. It is not clearly stated in the text how many of the 44 patient-derived organoids harbored HER2 alterations—specifically, how many had amplifications, mutations, or both. This breakdown should be provided.
4. Similarly, please clarify the HER2 status (mutation, amplification, or both) of the PDX-Os models tested in vivo with neratinib and T-DXd.
5. The conclusion that HER2 oncogenic dependence is not required for T-DXd sensitivity seems to be largely based on results from SMBO-106, which had HER2 1+ expression but also HER2 amplification (Fig. 2A). In the clinical data, only one HER2 1+ case is reported—and this patient also harbored a mutation. These findings may not be sufficient to support such conclusion and should be more cautiously framed.
6. The conclusion that ERBB2 co-mutation and amplification predicts exceptional clinical response is intriguing but based on a very small number of observations, and not fully supported in the manuscript by preclinical data. For instance, model #8, which harbors both mutation and amplification and was tested with T-DXd, is not discussed in the results. Including data from this model would strengthen the argument. Overall, I'd suggest describing this observation with more caution across the manuscript, given the limitations above.
7. A better description of the retrospective clinical cohort of patients with urothelial cancer treated with T-DXd is warranted in the results, so that the reader can better understand the context. Please provide key clinicopathologic details and the overall efficacy of the treatment in this cohort (e.g. rwPFS, OS).

Discussion:

8. The Discussion would benefit from a more logical and cohesive structure. Consider presenting the key findings in an order that reflects their progression—from molecular profiling and biological characterization of patient-derived organoids to in vivo sensitivity to T-DXd and the clinical data. In addition, the Discussion would be strengthened by better contextualizing the findings within the broader landscape of HER2 biology and T-DXd sensitivity in other solid tumors

MINOR:

Abstract

1. The statement that “urothelial cancers had the highest frequency of ERBB2 alterations” appears inconsistent with Figure 1A, which shows esophageal cancers (ESCA) with higher frequency. Please revise for accuracy.
2. We recommend specifying the total number of urothelial cancers (n=2,035) in the abstract and providing a breakdown of ERBB2 alterations. For example: “ERBB2 alterations were present in XX%: mutations (XX%), concurrent mutation and amplification (XX%), amplification alone (XX%)”
3. Instead of “urothelial cancer patients”, I'd recommend the use of the form “patients with urothelial cancer”, generally considered more respectful to patients.

Results:

4. Please clarify whether HER2 S310F/Y mutations are known activating alterations.
5. The statement that ERBB2 mutations/amplifications were largely mutually exclusive with FGFR3 mutations is interesting. We suggest discussing this further, as it may have therapeutic implications.
6. The reported 46% discordance in ERBB2 status between matched tumor-PDX pairs is noteworthy. Is this degree of discordance common in other solid tumors? Some context or comparison would be helpful.
7. Reverse Phase Protein Array is usually abbreviated as RPPA

Discussion:

8. The statement that T-DXd received tumor-agnostic accelerated approval for patients with HER2 IHC 3+ expression—based in part on the DESTINY-PanTumor02 trial—is repeated twice in the discussion. Consider removing the redundancy.

Reviewer #3

(Remarks to the Author)

The manuscript by Chen et al presents a deep but rather descriptive study regarding the frequency of ERBB2/HER2 alterations in urothelial carcinoma, and how these affect therapeutic response to HER2 inhibition with small molecules and a clinically-approved Trastuzumab ADC.

The major “handicap” of this study is that to a large extent (4/6 main figures) it covers the molecular characterization of the cohorts and the patient-derived tools developed. Figure 5 is the preclinical study and Figure 6 describes the real patient data. The clinical value of the findings is not clear. The lack of mechanistic insights will make it less appealing for the broad audience of NCOMMS. In fact, while the title implies that the authors identify the determinants of the sensitivity to HER2-ADCs, the study only concludes that HER2 status is not necessarily indicative of the response. Other determinants are not studied/identified (beyond the co-existence of ERBB2 mutations, which is not a robust finding).

The authors indicate that patients with concomitant mutation and overexpression at the protein level are the exceptional responders. However, mutant tumors correspond to 8.2%, of which less than 20% carry amplification. This implies that approximately 2% of the patients could benefit (based on the real-world data).

On the other hand, the authors claim (and their PDX-106 supports this) that patients could benefit from HER2-directed ADCs, independently of their HER2 expression status. Together with their finding that it is the ADC load that defines the response, most likely these “unexpected” responders are patients sensitive to topoisomerase inhibitors and not the ADC. This implies that in future clinical studies, the focus should be to identify the patients that are sensitive to this class of chemotherapy, irrespectively of their ERBB2/HER2 status. What mutations and/or epigenetic factors govern this favorable response is an interesting question on its own (e.g., sample 106 carries a BRIP1 stop codon which could imply homologous

recombination deficiency). The authors do not discuss this issue at all.

Specific points

1. The authors present data indicating that ERBB2/HER2 alterations correlate with tumor mutation burden, without providing any insight regarding the putative connection. On the other hand, the co-segregation of ERBB2/HER2 with TP53 and ERCC2 might explain the higher TMB which is likely attributed to TP53 and ERCC2 rather than ERBB2/HER2. There is relevant literature.
2. Besides the molecular characterization of the patient-derived tools, it is not clear what Figures 3 and 4 bring to the manuscript. For example, it is not clear what the meaning of clonal evolution is. Is this evolution that took place outside the patient? And if yes, what is the clinical relevance of this? Do the authors have the molecular profile of the starting material right out of the patient (primary culture or passage 0 graft)?
3. Figure 5 totally lacks statistical analysis. Which of the observed responses are statistically significant?
4. In Figure 6, the results can only be considered as indicative rather than conclusive due to the extremely low number of patients, particularly responders.

Reviewer #4

(Remarks to the Author)

The manuscript by Chen and Tang et al. describes multimodal analysis for investigating the effect of HER2-targeted ADC in urothelial cancer. Despite the high prevalence of HER2 gene abnormalities in urothelial cancers, the low therapeutic efficacy of HER2 antibodies and inhibitors is a major problem, and it is clinically important to clarify the association between the status of HER2 gene and sensitivity to HER2-targeted ADCs in urothelial cancers. The authors analyzed ERBB2 alterations in pan-cancer and showed the highest frequency of ERBB2 alterations in urothelial cancers. Using the bladder cancer PDX model, the authors showed that the status of ERBB2 was highly different between PDX and the original tumors. Also, HER2-targeted ADC was shown to be more effective than the HER kinase inhibitor in the PDX models. Clinically, cases with both mutations and amplifications were shown to be more effective for the HER2-targeted ADC.

This is a very large and interesting study, but it mostly analyzes bladder cancer and upper tract urothelial cancer together, and it is curious to know the results of the separate analysis, as there are various subtypes of each.

1. In Fig. 1, the authors have evaluated the mutation rate of ERBB2 in bladder cancer and UTUC together, but since there are differences in subtypes, they should be analyzed separately. At the very least, the separated data should be shown in the supplemental data.
2. In Fig. 1B, S310F/Y mutation was identified as the urothelial cancer predominant mutation. Do the authors explain the biological reason for the mutations that selectively occur in urothelial cancer?
3. In Fig. 2, 44 cases of PDX have been used, are these all derived from bladder cancer patients? It would be more informative to show the detailed information on PDX.
4. In Fig. S1E, at what time were the primary and metastatic cancer tissues collected? Usually, metastatic cancer is not sampled. Is it possible to assume that the primary and metastasis do not share the same ERBB2 mutation, which means that metastasis is occurring in the early stages of carcinogenesis?
5. Is the mismatch of ERBB2 mutation pairs in Figs 2A-2B tumor-PDXO pairs related to the mismatch of primary-metastasis tumor pairs? In some cases, HER2 mutation and amplification disappeared in PDXO. How to interpret this data?
6. SMBO-170 has too many ERBB2 copy numbers, compared to other tumors. Is it appropriate to include this kind of exceptional sample in the main analysis? Any specific reason for it?
7. In Fig.4, why is SMBO-114 luminal subtype but clearly positive for the basal marker KRT5 in Western blot?
8. In Fig. 6, it is curious to know whether patients with weakly positive HER2 in primary and HER2 positive in metastasis are resistant to T-Dxd?

Reviewer #5

(Remarks to the Author)

Version 1:

Reviewer comments:

Reviewer #1

(Remarks to the Author)
none

Reviewer #2

(Remarks to the Author)

All of our comments were adequately addressed. We would like to thank the authors for taking the time to revise the paper and to congratulate them for the excellent work.

Reviewer #3

(Remarks to the Author)

This is a much-improved manuscript. Based on all reviewers' comments, the authors rewrote sections of the manuscript, particularly the Discussion. Additional data were also provided. The sample size in real-world data increased, however conclusions regarding the efficacy of trastuzumab deruxtecan in patients and preclinical models are still drawn based on a small clinical cohort and few preclinical models. In this sense, my reservations regarding robustness of the findings remains. Although no mechanistic insight regarding sensitivity to HER ADCs is provided, the conclusions presented in this manuscript provide compelling evidence for a broader use of HER2 ADCs independently of HER2 expression status.

One minor point: TCGA is an acronym for "The Cancer Genome Atlas" and not "Tumor Cancer Genomic Atlas". Please correct throughout the manuscript.

Reviewer #4

(Remarks to the Author)

most of my original concerns were properly addressed by the authors.

Reviewer #5

(Remarks to the Author)

Dear Reviewers,

Please see below, in blue, a point-by-point response to each comment and suggestion. To address the comments and suggestions raised by the reviewers, we have added additional data to Figures 2, 3, 4, and 6 and Supplementary Figures 3, 5, and 6. We have also added one new table (Supplementary Table 1) and a new supplementary figure (Supplementary Figure 1). We believe that these changes have strengthened the manuscript and hope that the revised version will be acceptable to publication in *Nature Communications*.

Reviewer's comments:

Reviewer #1 (Remarks to the Author):

1. Can authors mention differential HER2 alterations based on upper tract vs bladder primary tumors?

We thank the reviewers for highlighting that bladder and upper tract urothelial cancers often have differences in the prevalence of oncogenic alterations and that a separate analysis of bladder and upper tract urothelial carcinoma (UTUC) could identify difference that would be of biologic and clinical importance. In response to the reviewer's suggestion, we have included in a revised supplementary figure (**Supplementary Figure 1**), a comparison of the frequency of *ERBB2* alterations in urothelial cancers arising in the bladder versus upper tract sites (**Supplementary Figure 1B**) as well as a lollipop plot comparing the distribution of missense mutations in *ERBB2* in bladder and upper tract cancers (**Supplementary Figure 1C**). These analyses revealed a higher frequency of *ERBB2* alterations in urothelial cancers arising in the bladder (bladder cancer cohort, 22.0%, n=1865) as compared to those tumors arising in an upper tract primary site (UTUC cohort, 16.5%, n=381). While the prevalence of *ERBB2* alterations is lower in UTUC, we believe that the 16.5% prevalence in this subset makes HER2 an attractive therapeutic target for patients with UTUC whose tumors exhibit *ERBB2* alterations. In addition to the revised **Supplementary Figure 1**, we have also highlighted these data in the text of the results section.

2. Is there an association between *ERBB2* mutations and *ERBB2* mRNA/protein expression?

In the initial submission, we analyzed the correlation between *ERBB2* amplification and HER2 protein expression using data generated as part of the urothelial and breast cancer cohorts of the Tumor Cancer Genomic Atlas. This analysis identified a weaker correlation between *ERBB2* amplification and protein expression in urothelial cancer (cor. = 0.69, $p < 0.001$) as compared to breast cancer (cor. = 0.80, $p < 0.001$). In response to the reviewer's comment, we have expanded this analysis to assess the relationship between *ERBB2* mutations and HER2 protein expression, which again revealed a weaker correlation in urothelial cancer (cor. = 0.72, $p < 0.001$) compared to breast cancer (cor. = 0.79, $p < 0.001$) (see figure below).

Based on the reviewer's question, we have revised the manuscript to include these analyses as part of an updated version of **Supplementary Figure 3C** (shown below), with the mutation and amplification status of individual samples color coded in a unified analysis of both.

3. The mechanism for sensitivity of HER2 mutations in particular to HER2 binding ADCs warrants some discussion. This appears similar to the phenomenon seen in NSCLC with HER2 mutations where greater internalization of the ADC appears to occur?

We agree with the reviewer that the sensitivity of tumors to antibody drug conjugates is complex and likely depends on an interplay between target expression, differences in the rate of antibody internalization, and sensitivity to the cytotoxic payload. Using isogenic models, members of our

research team have previously shown that cells with *ERBB2* mutations have a higher rate of antibody internalization, which may explain the clinical activity of T-DXd in lung cancer patients with activating HER2 mutations but low HER2 expression (Li et. al, 2020, PMID: 32213539).¹ The observation that HER2 mutation may enhance the rate of ADC internalization also provides a potential explanation as to the exceptional responses to T-DXd observed in patients whose tumors have both *ERBB2* amplification and mutation in our real-world cohort as such patients would be predicted to have both greater ADC target expression on the cancer cell surface and enhanced ADC internalization as compared to those patients with overexpression of wildtype HER2. Based on the reviewer's suggestion, we have revised the discussion to highlight that the higher rate of antibody internalization in the setting of HER2 mutation may explain in part the exceptional clinical responses observed in patients with both *ERBB2* mutation and amplification in our real-world cohort.

4. Another determinant of ADC activity is the linker, which has not been explored in this paper. This point could be discussed.

We thank the reviewer for this suggestion. We agree that differences in the linker among ADCs may impact their clinical activity. While we believe that studies comparing ADCs with different linkers would be beyond the scope of the current report, we have as suggested by the reviewer, revised the discussion to highlight that the high degree of *ERBB2* heterogeneity in urothelial cancers supports the use of ADCs with cleavable linkers which have the potential to overcome ADC cell surface target expression heterogeneity via the bystander effect.

5. While T-Dxd monotherapy has been approved after robust activity seen across tumors, the combination of T-Dxd and nivolumab was underwhelming in a 2nd line trial (Galsky et al)- please incorporate into discussion with potential reasons- or is this possibly a powering issue and sample size in these trials. In contrast disitamab vedotin + PD1 inhibition appears highly active, likely due to a tubulin toxin payload, which may be more optimal? Can the molecular data in this study be used to offer reasons why a tubulin toxin may be more optimal?

We agree with the reviewer that properties of ADCs such as the payload may influence their ability to synergize with immune checkpoint blockade. It has been hypothesized that the vedotin payload incorporated into enfortumab vedotin and disitamab vedotin may be more effective at inducing immunogenic cell death than the deruxtecan payload incorporated into T-DXd.² However, there is no published randomized data comparing ADCs with deruxtecan versus vedotin payloads for patients with urothelial cancer and we thus believe that these early clinical results should be interpreted with caution. Further, T-DXd in combination with pembrolizumab has shown encouraging efficacy in heavily pretreated patients with HER2+ and HER2-low metastatic breast cancer and may be more effective than either drug alone (DS8201-A-U106, NCT04042701). In sum, it is possible that the underwhelming results of T-DXd + nivolumab may have been attributable to differences in the patient population enrolled in the studies to date as highlighted by the reviewer.

Given the suggestion by the reviewer, we have revised the discussion to better address this ongoing area of investigation and the need for future studies to assess whether differences in ADC payload result in differences in synergy with immune checkpoint inhibitors. We also highlight in the revised discussion the need to develop immune competent models of *ERBB2* amplified and mutant urothelial cancer to allow for direct comparisons of the ability of HER2 ADCs with different payloads to enhance the antitumor activity of immune checkpoint inhibitors.

6. The potential reasons for lack of sensitivity of ERBB2 mutated tumors to TKIs could be explored in this dataset given comprehensive profiling data. Is this possible?

We agree with the reviewer that our patient derived models are a unique resource for understanding the disappointing results with HER2-targeted kinases inhibitors in patients with urothelial cancer and this was a focus of **Figure 5**. Specifically, in comparison to the breast cancer cell line BT-474, which has *ERBB2* amplification and has been shown previously to be sensitive to HER2 inhibition, higher concentrations of neratinib were required to inhibit downstream effectors of HER2 such as ERK and AKT in our bladder cancer organoid models, including SMBO-170, which has highest level *ERBB2* amplification in our organoid biobank. In SMBO-114, which harbors a *ERBB2* S310F hotspot mutation, treatment with neratinib was unable to suppress both ERK and AKT phosphorylation suggesting that activation of these downstream effectors is not dependent on HER2 in the SMBO-114 model. A notable finding from our studies of the organoid models was that neratinib did not inhibit the expression of pAKT in any of the five urothelial cancer models testing. As breast cancer cells with HER2 amplification have been shown previously to be selectively addicted to Akt signaling³, these data support our conclusion that *ERBB2* mutant and amplified urothelial cancer models do not display the same level of oncogenic addiction to HER2 as observed in breast cancer models. Given the reviewer's comment, we have revised this portion of the manuscript to better highlight these findings.

7. One interesting experiment may have been to explore activity of T-DXd following resistance to EV or even exploring the combination. These points could be discussed?

We agree with the reviewer that understanding whether urothelial cancers with acquired or intrinsic resistance to enfortumab vedotin exhibit cross-resistance to T-DXd or other HER2-targeted ADCs such as disitamab vedotin will be critical for guiding future clinical trials of novel ADCs. Unfortunately, the urothelial cancer patient derived models in the current study were generated before the widespread clinical use of EV. As highlighted in **Supplemental Figure 8**, our biobank does contain models such as SMBO-170 (HER2 amplified) which exhibited greater sensitivity to T-DXd than EV. These data indicate that a subset of patients with urothelial cancer may derive greater benefit from HER2-targeted ADCs than Nectin-4 targeted ADCs. Additionally, we believe these data support future clinical trials directly comparing HER2-targeted ADCs and Nectin-4 targeted ADCs in molecularly defined populations most likely to benefit from a HER2-targeted approach. To address the reviewer's suggestion, we have revised the discussion to emphasize that future studies should seek to develop novel patients-derived models from tumors collected post-EV treatment to study potential cross resistance with novel ADC therapies.

Reviewer #2 (Remarks to the Author):

This manuscript investigates the landscape and significance of ERBB2 alterations in urothelial cancers using a large cohort of 2,035 patients with tumor genomic profiling data. The authors develop and biologically characterize novel patient-derived bladder cancer models with HER2 alterations, and assess their in vivo sensitivity to trastuzumab deruxtecan (T-DXd). These preclinical findings are complemented by data from a real-world clinical cohort of 28 patients with metastatic urothelial cancer treated with T-DXd. This is a timely and relevant study that provides valuable insights into HER2 biology and therapeutic response to HER2-targeted drugs in urothelial cancer.

We offer the following comments and suggestions to help improve the quality of the manuscript:

MAJOR COMMENTS:

Introduction and Discussion:

1. Given that the manuscript addresses both HER2 amplification and mutations, we recommend broader discussion of the clinical data on HER2-mutant tumors treated with T-DXd. In particular, consider integrating findings from DESTINY-PanTumor01 (Li et al., Lancet Oncology, 2024), which includes 7 patients with HER2-mutant urothelial carcinoma.

As suggested by the reviewer, we have revised the manuscript to include additional discussion of recently reported clinical data with T-DXd pan-cancer including specific references to and discussion of the DESTINY-PanTumor01 (Li et al., Lancet Oncology, 2024) study⁴. As highlighted by the reviewer, this study included 7 patients with urothelial cancer. The ORR was 29.4% pan-cancer with 2 of 7 patients with urothelial cancer achieving an objective response. While these results were promising, the results suggested the need for further studies to define the role of T-DXd in patients with urothelial cancer whose tumors harbor *ERBB2* mutations, a goal of our real-world analyses in **Figure 6**. We thus believe that our current study provides further support for targeting HER2 with ADCs in patients with urothelial cancers, in particular, those with *ERBB2* mutation and amplification, points we have sought to better highlight in the revised discussion.

2. Although the approval and role of trastuzumab deruxtecan is widely covered, very little details are provided on disitamab vedotin, which has also shown impressive activity in HER2-altered urothelial cancer and is currently approved in China for this indication. We'd recommend a better description and discussion of this agent in the context of other anti-HER2 agents for urothelial cancer.

As suggested by the reviewer, we have added additional discussion of disitamab vedotin, a second HER2 targeted ADC which has also demonstrated promising clinical results in patients with urothelial cancer. In particular, and related to comment 5 above from Reviewer 1, we highlight in the revised discussion that the vedotin payload of distimab vedotin may induce greater immunogenic cell death than the deruxtecan payload of T-DXd and thus may be more effective than T-DXd when combined with immune checkpoint inhibitors. The availability of HER2-targeted ADCs with different cytotoxic payloads may also allow for sequential therapy for patients with acquired resistance driven by resistance to the cytotoxic payload as opposed to loss of target expression or impaired ADC internalization or trafficking, points we have sought to better highlight in the revised discussion.

Results:

3. It is not clearly stated in the text how many of the 44 patient-derived organoids harbored HER2 alterations—specifically, how many had amplifications, mutations, or both. This breakdown should be provided.

We thank the reviewer for this comment. To address this suggestion, we have added a new **Supplementary Table 1** that provides additional information on the patient derived models including whether the model was derived from a bladder or UTUC primary site, the patients initial stage, tumor grade, pathologic classification and the sex of the patient. The *ERBB2* mutational status of the patient derived models and the tumors from which they were derived is also included in the OncoPrint in **Figure 2A**. Additionally, to assist the reader, we have added annotation as to the *ERBB2* mutational status of the models studied in **Figure 2D** and **Figure 5A** to the respective figure panels.

4. Similarly, please clarify the HER2 status (mutation, amplification, or both) of the PDX-Os models tested in vivo with neratinib and T-DXd.

To address the reviewer's suggestion, we now highlight the *ERBB2* mutational status of the patient derived models in **Figures 5A** and **Figure 5C**. Consistent with the lower correlation between *ERBB2* amplification status and HER2 protein expression noted in the TCGA bladder versus breast cohorts, SMBO-106, while derived from a tumor classified clinically as amplified (total *ERBB2* copy number of 9 by next generation sequencing), had relatively low levels of HER2 expression similar to those of non-amplified models as highlighted by the western blot panel in **Figure 2D**.

5. The conclusion that HER2 oncogenic dependence is not required for T-DXd sensitivity seems to be largely based on results from SMBO-106, which had HER2 1+ expression but also HER2 amplification (Fig. 2A). In the clinical data, only one HER2 1+ case is reported—and this patient also harbored a mutation. These findings may not be sufficient to support such conclusion and should be more cautiously framed.

A notable result of the clinical trials of T-DXd in patients with breast cancer is the surprising degree of clinical activity of this HER2-targeted ADC in patients resistant to older HER2-targeted therapies such as kinase inhibitors and the clinical activity of T-DXd in patients with low or even ultra-low levels of HER2 expression. These results suggest that in contrast to kinase inhibitors, oncogenic dependence on HER2 is not required for T-DXd sensitivity in breast cancer. Our analyses of the HER kinase inhibitor neratinib using our urothelial cancer organoid models revealed that they lack the HER2-dependence of HER2 amplified breast cancer cells. While neratinib fails to inhibit downstream signaling effectors of HER2 such as ERK and AKT and does not induce apoptosis of SMBO-106 organoids, SMBO-106 xenografts were exquisitely sensitive to T-DXd. These data suggest that similar to breast cancer, sensitivity to T-DXd in a urothelial cancer context does not require HER2 oncogene dependence and support broader testing of HER2-targeted ADCs in urothelial cancer including patients with low levels of HER2 expression, as has now been successfully pursued in patients with breast cancer. Given the reviewer's comments, we have sought to revise the results and discussion sections to better highlight the limitations of our current data and the need for future studies to better define the activity of T-DXd in patients with urothelial cancer whose tumor have low HER2 expression.

6. The conclusion that *ERBB2* co-mutation and amplification predicts exceptional clinical response is intriguing but based on a very small number of observations, and not fully supported in the manuscript by preclinical data. For instance, model #8, which harbors both mutation and amplification and was tested with T-DXd, is not discussed in the results. Including data from this model would strengthen the argument. Overall, I'd suggest describing this observation with more caution across the manuscript, given the limitations above.

To clarify, our conclusion that co-amplification and mutation of *ERBB2* may be a predictor of exceptional response was based on our real-world clinical cohort and not the preclinical studies of the patient derived organoids. While our organoid biobank is among the largest reported to date for urothelial cancer, the number of models with *ERBB2* mutation is too small in our opinion to conclusively determine the role of individual mutations such as *ERBB2* in dictating drug sensitivity given the diverse co-mutation patterns observed in urothelial cancer. To address this comment by Reviewer 2, we have expanded our real-world cohort of patients from 28 to 40 by including additional patients treated with T-DXd after our prior study cutoff and by updating the clinical data of the patients previously included in the initial manuscript. These updated clinical

data indicate that 3 of 4 patients with the longest duration of response to T-DXd had both amplification and mutation. While we agree these data are hypothesis generating, they strongly support future clinical studies to determine whether this genomically definable subset of patients represent those most likely to achieve durable responses to T-DXd. To address the reviewer's concerns, we have also sought to better highlight the limitations of our cohort and the need for future studies in the discussion. We have also updated the manuscript and figures to include additional preclinical data on SCBO-8 including single cell DNA (**Fig. 3, Supplementary Fig. 5**) and scRNA seq (**Fig. 4, Supplementary Fig. 6**) analyses of SCBO-8.

7. A better description of the retrospective clinical cohort of patients with urothelial cancer treated with T-DXd is warranted in the results, so that the reader can better understand the context. Please provide key clinicopathologic details and the overall efficacy of the treatment in this cohort (e.g. rwPFS, OS).

We thank the reviewer for this suggestion. To address the reviewer's comment, we have now added an OncoPrint and Kaplan-Meier curve to **Figure 6** to provide a clearer overview of co-mutational status of the tumors of patients treated with T-DXd and the progression free survival of this cohort.

Discussion:

8. The Discussion would benefit from a more logical and cohesive structure. Consider presenting the key findings in an order that reflects their progression—from molecular profiling and biological characterization of patient-derived organoids to in vivo sensitivity to T-DXd and the clinical data. In addition, the Discussion would be strengthened by better contextualizing the findings within the broader landscape of HER2 biology and T-DXd sensitivity in other solid tumors.

As suggested by the reviewer, we have extensively revised the discussion to better highlight the key findings of this manuscript with a focus on better contextualizing the findings in relationship to the broader landscape of data on T-DXd and HER2-targeted therapies pan-cancer.

MINOR:

Abstract

1. The statement that “urothelial cancers had the highest frequency of ERBB2 alterations” appears inconsistent with Figure 1A, which shows esophageal cancers (ESCA) with higher frequency. Please revise for accuracy.

We thank the reviewer for pointing out this error. We had sought to imply that oncogenic *ERBB2* mutations (missense mutations and small indels) were more common in urothelial cancers (not including amplification) than in other solid tumors but agree that the language used was imprecise as gene amplification is a type of mutation. We have thus amended the manuscript for clarity to address the reviewer's comment.

2. We recommend specifying the total number of urothelial cancers (n=2,035) in the abstract and providing a breakdown of ERBB2 alterations. For example: “ERBB2 alterations were present in XX%: mutations (XX%), concurrent mutation and amplification (XX%), amplification alone (XX%)”

We thank the reviewer for this suggestion and have updated the abstract and results sections to specify the total number of urothelial cancers analyzed and provide a clearer breakdown of *ERBB2* alterations as recommended. Specifically, we now state: “In a cohort of 42,415

prospectively analyzed solid tumors, 14.5% of urothelial cancers (n=295/2,035) had *ERBB2* alterations (6.7% *ERBB2* mutation, 6.3% amplification of wildtype *ERBB2*, and 1.5% concurrent mutation and amplification).

3. Instead of “urothelial cancer patients”, I’d recommend the use of the form “patients with urothelial cancer”, generally considered more respectful to patients.

We agree with the reviewer and have revised the manuscript to replace “urothelial cancer patients” with “patients with urothelial cancer” throughout the text.

Results:

4. Please clarify whether HER2 S310F/Y mutations are known activating alterations.

ERBB2 S310F/Y mutation has been previously shown using isogenic models to activate downstream signaling pathways by promoting noncovalent receptor dimerization and kinase activation (Greulich et al., 2012, PMID: 22908275)⁵. S310F/Y was also shown to be oncogenic in a breast cancer context (Bose et al., 2013, PMID: 23220880)⁶. To address the reviewer’s question, we have referenced these data in the revised manuscript.

5. The statement that *ERBB2* mutations/amplifications were largely mutually exclusive with *FGFR3* mutations is interesting. We suggest discussing this further, as it may have therapeutic implications.

We thank the reviewer for highlighting the mutual exclusivity of *ERBB2* and *FGFR3* mutation. To address the reviewer’s comment, we further analyzed the 33 patients with urothelial cancer in our cohort who had both *ERBB2* amp/mut and *FGFR3* mutation. *FGFR3* mutations were clonal in 56% of these cases, sub-clonal in 20.6%, and indeterminate in 23.5% (see Figure below), indicating that *ERBB2* and *FGFR3* alterations may represent distinct populations of tumor cells in a subset of tumors with mutations in both. These data suggest that there may be a subset of urothelial cancers in which *ERBB2* and *FGFR3* mutations co-exist but in different cancer cells. While we believe this observation warrants further study, potentially using emerging single cell DNA sequencing methods, we believe that such studies are beyond the scope of the current report.

6. The reported 46% discordance in ERBB2 status between matched tumor-PDX pairs is noteworthy. Is this degree of discordance common in other solid tumors? Some context or comparison would be helpful.

We thank the reviewer for highlighting this point as we believe it is a particularly notable finding of the manuscript. While definitive data is lacking for most cancer types, our experience suggests that the prevalence of *ERBB2* discordance between primary and metastatic disease sites varies as a function of cancer type. *ERBB2* heterogeneity has also been found to be common in esophagogastric cancers where it has been shown to be a putative mechanism of resistance to HER2 targeted therapies.^{7,8} Conversely, HER2 amplification has been reported to be largely concordant in breast cancer primary and patient-matched metastases.⁹ Based on the reviewer's comment, we have added further discussion related to the discordance of *ERBB2* status comparing urothelial to other solid tumors in the revised Discussion section and its potential role in mediating resistance to HER2-targeted therapies.

7. Reverse Phase Protein Array is usually abbreviated as RPPA

We thank the reviewer for pointing out this typo and have corrected it in the revised manuscript.

Discussion:

8. The statement that T-DXd received tumor-agnostic accelerated approval for patients with HER2 IHC 3+ expression—based in part on the DESTINY-PanTumor02 trial—is repeated twice in the discussion. Consider removing the redundancy.

As suggested by the reviewer, we have removed this redundancy in the revised manuscript.

Reviewer #3 (Remarks to the Author):

The manuscript by Chen et al presents a deep but rather descriptive study regarding the frequency of ERBB2/HER2 alterations in urothelial carcinoma, and how these affect therapeutic

response to HER2 inhibition with small molecules and a clinically-approved Trastuzumab ADC. The major “handicap” of this study is that to a large extent (4/6 main figures) it covers the molecular characterization of the cohorts and the patient-derived tools developed. Figure 5 is the preclinical study and Figure 6 describes the real patient data. The clinical value of the findings is not clear. The lack of mechanistic insights will make it less appealing for the broad audience of NCOMMS. In fact, while the title implies that the authors identify the determinants of the sensitivity to HER2-ADCs, the study only concludes that HER2 status is not necessarily indicative of the response. Other determinants are not studied/identified (beyond the co-existence of ERBB2 mutations, which is not a robust finding).

The authors indicate that patients with concomitant mutation and overexpression at the protein level are the exceptional responders. However, mutant tumors correspond to 8.2%, of which less than 20% carry amplification. This implies that approximately 2% of the patients could benefit (based on the real-world data).

On the other hand, the authors claim (and their PDX-106 supports this) that patients could benefit from HER2-directed ADCs, independently of their HER2 expression status. Together with their finding that it is the ADC load that defines the response, most likely these “unexpected” responders are patients sensitive to topoisomerase inhibitors and not the ADC. This implies that in future clinical studies, the focus should be to identify the patients that are sensitive to this class of chemotherapy, irrespectively of their ERBB2/HER2 status. What mutations and/or epigenetic factors govern this favorable response is an interesting question on its own (e.g., sample 106 carries a BRIP1 stop codon which could imply homologous recombination deficiency). The authors do not discuss this issue at all.

Reviewer’s Comments:

In fact, while the title implies that the authors identify the determinants of the sensitivity to HER2-ADCs, the study only concludes that HER2 status is not necessarily indicative of the response. Other determinants are not studied/identified (beyond the co-existence of ERBB2 mutations, which is not a robust finding).

As highlighted by the reviewer, a main conclusion of our preclinical studies was that the sensitivity of T-DXd was not restricted to tumors with high HER2 protein overexpression in urothelial cancer. Our data also suggest that sensitivity to the cytotoxic payload, in particular its ability to induce cancer cell death, may be more predictive of clinical response to T-DXd than HER2 mutational or expression status, supporting broader use of T-DXd in patients with urothelial cancer than that covered by current tumor agnostic FDA authorization. Our results also support the development of HER2 ADCs with novel cytotoxic payloads, in particular those that could enhance immunotherapy response and functional precision oncology diagnostic platforms capable of quantitating payload sensitivity pre-treatment thus allowing for greater individualization of therapy. Given the reviewer’s comment, we have sought to revise the discussion to better highlight these points.

The authors indicate that patients with concomitant mutation and overexpression at the protein level are the exceptional responders. However, mutant tumors correspond to 8.2%, of which less than 20% carry amplification. This implies that approximately 2% of the patients could benefit (based on the real-world data).

We agree with the reviewer that urothelial cancers with concurrent amplification and mutation are rare but the clinical experience with targeted therapies indicates that targeting molecularly defined subsets of patients with low frequency alterations such as ALK-fusion or BRAF mutant lung cancers can have significant clinical impact. As both Nectin-4 and HER2-targeted ADCs have now demonstrated compelling clinical activity in patients with urothelial cancer, a major clinical challenge will be to identify biomarkers that could help identify those patients most likely

to benefit from HER2 versus Nectin-4 targeted therapies. Our data suggesting that concurrent mutation and amplification of HER2 may represent a biomarker of exceptional/durable response to HER2-targeted ADCs provides justification for future clinical studies using pretreatment biomarkers to select between HER2 and Nectin-4-targeted therapies and potentially other ADCs in early clinical development.

On the other hand, the authors claim (and their PDX-106 supports this) that patients could benefit from HER2-directed ADCs, independently of their HER2 expression status. Together with their finding that it is the ADC load that defines the response, most likely these “unexpected” responders are patients sensitive to topoisomerase inhibitors and not the ADC. This implies that in future clinical studies, the focus should be to identify the patients that are sensitive to this class of chemotherapy, irrespectively of their ERBB2/HER2 status. What mutations and/or epigenetic factors govern this favorable response is an interesting question on its own (e.g., sample 106 carries a BRIP1 stop codon which could imply homologous recombination deficiency). The authors do not discuss this issue at all.

We agree with the reviewer that sensitivity to ADCs is complex with both expression/mutation of the cell surface target and payload sensitivity influencing the likelihood of patient response. In fact, our integrated preclinical and clinical data suggest that both HER2 mutational status and payload sensitivity are both key determinants of response as highlighted by the reviewer. While we also agree that individual co-mutations in genes such as BRIP1 could prove to be clinically relevant molecular determinants of ADC response, given the complex co-mutation patterns characteristic of human bladder cancers, we believe that such a determination would require the generation of isogenic models for each mutation of interest and thus would be beyond the scope of the current manuscript.

To address the reviewers suggestion, we have expanded the discussion to the degree possible given space constraints to better highlight how future studies should seek to expand the available ADC payloads and the need to develop diagnostic platforms and biomarkers that can be used pre-treatment to guide clinicians in individualizing treatment by choosing the ADC with the payload most likely to be effective for an individual patients. As HER2-targeted ADCs with both deruxtecan and vedotin have demonstrated compelling clinical responses, the need to choose among ADCs that target the same cell surface protein but have different cytotoxic payloads is now an urgent clinical need. More specifically, we conclude the discussion with the statement: “the exquisite sensitivity of SMBO-106 (low HER2 expression) xenografts to T-DXd which mirrored the high degree of sensitivity of SMBO-106 organoids to exatecan-induced cell death suggests that the cytotoxic payload may be a more critical predictor of HER2-targeted ADCs response than HER2 expression levels, supporting ongoing efforts to develop functional precision oncology diagnostic platforms capable of quantitating payload sensitivity pre-treatment thus allowing for greater individualization of therapy.”

Other Specific points

1. The authors present data indicating that ERBB2/HER2 alterations correlate with tumor mutation burden, without providing any insight regarding the putative connection. On the other hand, the co-segregation of ERBB2/HER2 with TP53 and ERCC2 might explain the higher TMB which is likely attributed to TP53 and ERCC2 rather than ERBB2/HER2. There is relevant literature.

We thank the reviewer for highlight the potential correlation between tumor mutational burden (TMB) and *TP53* and *ERCC2* mutational status. While we agree that a more in-depth study of

the role of *ERCC2* and *TP53* mutation are important, we believe that they would be best pursued as part of dedicated future studies. In this respect, we believe that the models generated and characterized in the current report will be a resource for the broader research community and we are committed to making all models available for research use. To address the reviewer's comment, we also now include an OncoPrint in the revised **Figure 6** demonstrating that responses to T-DXd were observed in both *TP53* mutant and wildtype patients.

2. Besides the molecular characterization of the patient-derived tools, it is not clear what Figures 3 and 4 bring to the manuscript. For example, it is not clear what the meaning of clonal evolution is. Is this evolution that took place outside the patient? And if yes, what is the clinical relevance of this? Do the authors have the molecular profile of the starting material right out of the patient (primary culture or passage 0 graft)?

One of the notable findings of our study is the high degree of discordance in *ERBB2* mutational status between primary and metastatic disease sites as well as discordance between the patient derived models and the tumors from which they were derived. A potential advantage or short-term organoid models over older 2D cell lines that have been passaged for years in culture is that they potential retain at least some of the molecular heterogeneity characteristic of human cancers. Our goal in **Figures 3 and 4** was to assess the heterogeneity of select HER2 altered organoid models at the single cell level using both single cell DNA and RNA profiling methods. Our single cell RNA sequencing data indicated that all the models profiled exhibited HER2 expression heterogeneity, and we believe that this molecular heterogeneity which is similar to that observed in patients with urothelial cancer will make them useful research tools for the broader scientific community. Additionally, our single cell DNA sequencing data revealed that in the case of SMBO-109, a model with *ERBB2* discordance between the patient-derived organoid and the tumor from which it was derived that we also observed heterogeneity of *ERBB2* amplification with two distinct populations, one with 11 copies of *ERBB2* and the other with 17 copies. In the revised manuscript, we have sought to better highlight the potential clinical significance of this *ERBB2* mutational and expression heterogeneity. To facilitate additional research in the field, we are also committed as above to making all models described in the current report available as a resource to the scientific community.

3. Figure 5 totally lacks statistical analysis. Which of the observed responses are statistically significant?

As suggested by the reviewer, we have amended the figures to add additional statistical analyses to the functional studies in **Figure 5**, and **Supplementary Figures S7 and S8**. Significance notations are also now included in the updated figure and legends.

4. In Figure 6, the results can only be considered as indicative rather than conclusive due to the extremely low number of patients, particularly responders.

We agree with the reviewer that the conclusions from **Figure 6** are hypothesis generating. To address the reviewer's concern, we have added additional patients to our real-world cohort of patients with urothelial cancer treated with T-DXd and additional tumor molecular profiling data to **Figure 6** to facilitate future studies combining this cohort with other studies of patients with urothelial cancer treated with T-DXd. We also believe that the data presented justify future studies testing whether co-mutation and amplification of *ERBB2* identify a subset of patients with urothelial cancer most likely to achieve durable responses and studies of pre-existent *ERBB2* mutational and HER2 expression heterogeneity as a predictor of early disease

progression. To address the reviewer's comment, we have also sought to better outline how the current study can be used to guide future studies of HER2-targeted ADCs as part of the revised discussion.

Reviewer #4 (Remarks to the Author):

The manuscript by Chen and Tang et al. describes multimodal analysis for investigating the effect of HER2-targeted ADC in urothelial cancer. Despite the high prevalence of HER2 gene abnormalities in urothelial cancers, the low therapeutic efficacy of HER2 antibodies and inhibitors is a major problem, and it is clinically important to clarify the association between the status of HER2 gene and sensitivity to HER2-targeted ADCs in urothelial cancers. The authors analyzed *ERBB2* alterations in pan-cancer and showed the highest frequency of *ERBB2* alterations in urothelial cancers. Using the bladder cancer PDX model, the authors showed that the status of *ERBB2* was highly different between PDX and the original tumors. Also, HER2-targeted ADC was shown to be more effective than the HER kinase inhibitor in the PDX models. Clinically, cases with both mutations and amplifications were shown to be more effective for the HER2-targeted ADC.

This is a very large and interesting study, but it mostly analyzes bladder cancer and upper tract urothelial cancer together, and it is curious to know the results of the separate analysis, as there are various subtypes of each.

1. In Fig. 1, the authors have evaluated the mutation rate of *ERBB2* in bladder cancer and UTUC together, but since there are differences in subtypes, they should be analyzed separately. At the very least, the separated data should be shown in the supplemental data.

We thank the reviewers for highlighting that bladder and upper tract urothelial cancers often have differences in the prevalence of oncogenic alterations and that a separate analysis of bladder and upper tract urothelial carcinoma (UTUC) could identify difference that would be of biologic and clinical importance. In response to the reviewer's suggestion, we have included in a revised supplementary figure (**Supplementary Figure 1**), a comparison of the frequency of *ERBB2* alterations in urothelial cancers arising in the bladder versus upper tract sites (**Supplementary Figure 1B**) as well as a lollipop plot comparing the distribution of missense mutations in *ERBB2* in bladder and upper tract cancers (**Supplementary Figure 1C**). These analyses revealed a higher frequency of *ERBB2* alterations in urothelial cancers arising in the bladder (bladder cancer cohort, 22.0%, n=1865) as compared to those tumors arising in an upper tract primary site (UTUC cohort, 16.5%, n=381). While the prevalence of *ERBB2* alterations is lower in UTUC, we believe that the 16.5% prevalence in this subset makes HER2 an attractive therapeutic target for patients with UTUC whose tumors exhibit *ERBB2* alterations. In addition to the revised **Supplementary Figure 1**, we have also highlighted these data in the results section.

2. In Fig. 1B, S310F/Y mutation was identified as the urothelial cancer predominant mutation. Do the authors explain the biological reason for the mutations that selectively occur in urothelial cancer?

We thank the reviewer for raising this interesting question. To address the reviewer's question, we amended the manuscript to cite prior data showing that HER2 S310F is an APOBEC-associated mutation, as described previously by Shi et al., 2020 PMID: 32988402.¹⁰ The reviewer's question also prompted us to perform an analysis showing that in our dataset the

S310F is more common in tumors with a predominant APOBEC signature (new **Supplementary Figure 1D**).¹⁰ These data provide a biologic explanation as to why HER2 S310F is particularly common in urothelial cancers, as APOBEC is the dominant mutational signature in most urothelial cancers (73.4% of our cohort).

3. In Fig. 2, 44 cases of PDX have been used, are these all derived from bladder cancer patients? It would be more informative to show the detailed information on PDX.

To address this suggestion by Reviewer 4, we have now included clinical information for each PDX and PDO model in new **Supplementary Table 1**. We agree with the reviewer that inclusion of this additional clinical annotation will make our organoid biobank a more useful resource for the broader research community.

4. In Fig. S1E, at what time were the primary and metastatic cancer tissues collected? Usually, metastatic cancer is not sampled. Is it possible to assume that the primary and metastasis do not share the same *ERBB2* mutation, which means that metastasis is occurring in the early stages of carcinogenesis?

We thank the reviewer for highlighting what we believe is a notable finding from our study, specifically, the high degree of *ERBB2* mutational discordance between patient-matched primary and metastatic tumor pairs and the urothelial cancer organoid models and the tumors from which they were derived. These findings suggest that *ERBB2* amplification and mutation is often a later event in urothelial cancer pathogenesis and not present in all tumor cells, which contrasts to breast cancer, a disease in which *ERBB2* amplification has been shown to be an early initiating event and highly concordant between primary and metastatic disease sites.¹¹ The results also suggest similarities between urothelial and esophagogastric cancers, a cancer type in which selection for *ERBB2* wildtype subclones has been shown to be a common mechanism of resistance to HER2-directed therapies.^{12,13} Together with the patient highlighted in **Figure 6D** who had rapid progression of disease on T-DXd and discordance of HER2 status between the pre-treatment tumor and the post-treatment sample, we believe that the data suggest that HER2 heterogeneity may be a common mechanism of resistance to HER2-targeted ADCs in patients with urothelial cancer. The results also suggest that molecularly profiling of patients with urothelial cancer should optimally use a new tumor sample collected immediately prior to starting a new therapy. Unfortunately, such a paradigm is often not feasible in the clinic given the invasive nature of tumor biopsies and cost and risks involved. Given the reviewer's comments, we have revised the discussion to better address these points.

5. Is the mismatch of *ERBB2* mutation pairs in Figs 2A-2B tumor-PDXO pairs related to the mismatch of primary-metastasis tumor pairs? In some cases, HER2 mutation and amplification disappeared in PDXO. How to interpret this data?

We agree that the mismatch observed in *ERBB2* status between tumor-PDX-O pairs is a consequence of the sub-clonal nature of *ERBB2* alterations in a subset of patients with urothelial cancer and thus linked to the high degree of discordance in *ERBB2* alterations between primary and metastatic tumor sites. We thank the reviewer for highlighting this finding which we believe is both biologically and clinically significant as discussed above in response to Question 4. As we have sought to better highlight in the revised discussion, our findings are analogous to the *ERBB2* heterogeneity reported in esophagogastric cancers which has been associated with resistance to HER2-targeted therapies (Sanchez-Vega et al., 2019, PMID: 30463996).¹² The patient highlighted in **Figure 6D** who had HER2 3+ expression pre-treatment with T-DXd but no HER2 expression in a metastasis collected following disease progression

also suggests that selection for tumor cells with lower HER2 expression or lacking *ERBB2* mutation/amplification may be a recurrent mechanism of resistance to HER2-targeted therapies in patients with urothelial cancer. A clinical implication of these findings is that analysis of archival tumor tissue collected from the primary tumor site may provide incorrect information as to the HER2 expression/*ERBB2* mutational status of metastatic sites and provides rationale for repeat molecular testing of a metastatic site or cfDNA as a guide to systemic therapy selection in patients with metastatic progression. Given the reviewer's comment, we have sought to improve the discussion to better address these points in the revised manuscript.

6. SMBO-170 has too many *ERBB2* copy numbers, compared to other tumors. Is it appropriate to include this kind of exceptional sample in the main analysis? Any specific reason for it?

We agree with the reviewer that urothelial cancers with very high levels of *ERBB2* copy number gain represent a minority of urothelial cancers with only 2.93% of the urothelial cancers in our cohort having >20 copies of the *ERBB2* gene. While only a minority of urothelial cancers, we do believe that this molecularly defined subset is of potential clinical significance. Rare genotypes have proven to be viable drug targets in other cancer types. For example, BRAF V600E is found in <2% of lung adenocarcinomas, but BRAF targeted therapies are now standard of care for this molecularly defined subset of patients with lung cancer.

To address the reviewer's comment, we have sought to better highlight in the revised manuscript that the level of *ERBB2* amplification observed in SMBO-170 is more common in breast and esophageal cancer and we believe that this difference may explain in part the greater clinical activity of older drugs such as trastuzumab and neratinib in these other cancer types. More specifically, as is now shown in new **Supplementary Fig. 1E**, 2.93% (113/3855) of bladder tumors in our cohort had >20 copies of *ERBB2* based on integer copy number analysis of our NGS data, compared to 7.06% (719/10182) of breast cancers and 10% (269/2689) of esophagogastric cancers. We believe that these data together with the less robust correlation between *ERBB2* amplification and HER2 protein expression in the bladder versus breast cancer TCGA cohorts as shown in **Supplementary Figure 3C**, suggest that the cutoff used to define clinically relevant *ERBB2* amplification in urothelial cancers, which was adapted from that used for esophagogastric cancers, may be suboptimal for use in patients with urothelial cancer. Given the reviewer's comment, we have sought to improve the presentation and discussion of these data in the revised manuscript.

7. In Fig.4, why is SMBO-114 luminal subtype but clearly positive for the basal marker KRT5 in Western blot?

We thank the reviewer for highlighting this observation. SMBO-114 was classified as luminal subtype based on analysis of the single cell RNA sequencing (scRNA-seq) data using the consensus classification schema as defined by Kamoun et al., 2020, PMID: 31563503.¹⁴ It is not uncommon for a subset of luminal bladder tumors as defined by transcriptional profiling to retain expression of some basal markers. To clarify and strengthen this point, we have added additional basal markers (shown in red: KRT5, CD44, KRT14) and luminal markers (shown in blue: Nectin-4 and GATA3) to **Figure 2D** of the revised manuscript highlighting the heterogeneous expression of luminal and basal markers across the organoid models. While the clinical implications of this observation will require further study, our experience suggests that luminal and basal classifications are not binary but rather represent a spectrum of phenotypes with some tumors exhibiting both luminal and basal features. As clinical response to ADCs such as enfortumab vedotin has been shown to correlate with luminal and basal phenotypes, the finding of tumors with features of both likely has important clinical implications.

8. In Fig. 6, it is curious to know whether patients with weakly positive HER2 in primary and HER2 positive in metastasis are resistant to T-DXd?

We agree with the reviewer that this will be an important question for future studies. Unfortunately, we are not aware of any large collection of paired primary and metastatic tumors collected pre-treatment from patients with urothelial cancer treated with T-DXd or other HER2-targeted ADCs that could be used to answer this question in a timely manner. Given the frequent discordance between HER2 status in primary and metastatic tumor samples identified in the current analysis, we have begun to suggest a biopsy at metastatic progression for our patients with urothelial cancer as a guide to systemic therapy treatment. Our hope is that in the future, a large cohort of paired primary and metastatic samples will be available to allow for the analysis proposed by the reviewer.

Reviewer #5 (Remarks to the Author):

We thank Reviewer 5 for their time co-reviewing this manuscript.

References

1. Li, B.T. *et al.* HER2-Mediated Internalization of Cytotoxic Agents in ERBB2 Amplified or Mutant Lung Cancers. *Cancer Discov* **10**, 674-687 (2020).
2. Muller, P. *et al.* Microtubule-depolymerizing agents used in antibody-drug conjugates induce antitumor immunity by stimulation of dendritic cells. *Cancer Immunol Res* **2**, 741-55 (2014).
3. She, Q.B. *et al.* Breast tumor cells with PI3K mutation or HER2 amplification are selectively addicted to Akt signaling. *PLoS One* **3**, e3065 (2008).
4. Li, B.T. *et al.* Trastuzumab deruxtecan in patients with solid tumours harbouring specific activating HER2 mutations (DESTINY-PanTumor01): an international, phase 2 study. *Lancet Oncol* **25**, 707-719 (2024).
5. Greulich, H. *et al.* Functional analysis of receptor tyrosine kinase mutations in lung cancer identifies oncogenic extracellular domain mutations of ERBB2. *Proc Natl Acad Sci U S A* **109**, 14476-81 (2012).
6. Bose, R. *et al.* Activating HER2 mutations in HER2 gene amplification negative breast cancer. *Cancer Discov* **3**, 224-37 (2013).
7. Cho, E.Y. *et al.* Heterogeneity of ERBB2 in gastric carcinomas: a study of tissue microarray and matched primary and metastatic carcinomas. *Mod Pathol* **26**, 677-84 (2013).
8. Augustin, J.E., Soussan, P. & Bass, A.J. Targeting the complexity of ERBB2 biology in gastroesophageal carcinoma. *Ann Oncol* **33**, 1134-1148 (2022).
9. Carlsson, J. *et al.* HER2 expression in breast cancer primary tumours and corresponding metastases. Original data and literature review. *Br J Cancer* **90**, 2344-8 (2004).

10. Shi, M.J. *et al.* Identification of new driver and passenger mutations within APOBEC-induced hotspot mutations in bladder cancer. *Genome Med* **12**, 85 (2020).
11. Razavi, P. *et al.* The Genomic Landscape of Endocrine-Resistant Advanced Breast Cancers. *Cancer Cell* **34**, 427-438 e6 (2018).
12. Sanchez-Vega, F. *et al.* EGFR and MET Amplifications Determine Response to HER2 Inhibition in ERBB2-Amplified Esophagogastric Cancer. *Cancer Discov* **9**, 199-209 (2019).
13. Maron, S.B. *et al.* Determinants of Survival with Combined HER2 and PD-1 Blockade in Metastatic Esophagogastric Cancer. *Clin Cancer Res* **29**, 3633-3640 (2023).
14. Kamoun, A. *et al.* A Consensus Molecular Classification of Muscle-invasive Bladder Cancer. *Eur Urol* **77**, 420-433 (2020).

Dear Reviewers,

We appreciate the constructive feedback provided during the review process, which has helped strengthen the manuscript. Please see below, in blue, a point-by-point response to each comment and suggestion.

Reviewer's comments:

Reviewer #1 (Remarks to the Author):

none

We thank the reviewer for their time and consideration.

Reviewer #2 (Remarks to the Author):

All of our comments were adequately addressed. We would like to thank the authors for taking the time to revise the paper and to congratulate them for the excellent work.

We thank the reviewer for the positive feedback and kind words.

Reviewer #3 (Remarks to the Author):

This is a much-improved manuscript. Based on all reviewers' comments, the authors rewrote sections of the manuscript, particularly the Discussion. Additional data were also provided. The sample size in real-world data increased, however conclusions regarding the efficacy of trastuzumab deruxtecan in patients and preclinical models are still drawn based on a small clinical cohort and few preclinical models. In this sense, my reservations regarding robustness of the findings remains. Although no mechanistic insight regarding sensitivity to HER ADCs is provided, the conclusions presented in this manuscript provide compelling evidence for a broader use of HER2 ADCs independently of HER2 expression status.

One minor point: TCGA is an acronym for "The Cancer Genome Atlas" and not "Tumor Cancer Genomic Atlas". Please correct throughout the manuscript.

We thank the reviewer for the positive feedback on the revised manuscript. We have corrected "Tumor Cancer Genomic Atlas" to "The Cancer Genome Atlas (TCGA)" throughout the manuscript.

Reviewer #4 (Remarks to the Author):

Most of my original concerns were properly addressed by the authors.

We thank the reviewer for the time and consideration.

Reviewer #5 (Remarks to the Author):

I co-reviewed this manuscript with one of the reviewers who provided the listed reports. This is

part of the Nature Communications initiative to facilitate training in peer review and to provide appropriate recognition for Early Career Researchers who co-review manuscripts.

We thank the reviewer for co-reviewing this manuscript.